# REVISITING INVERSE HESSIAN VECTOR PRODUCTS FOR CALCULATING INFLUENCE FUNCTIONS

## ABSTRACT

Influence functions are a popular tool for attributing models' outputs to training data. The traditional approach relies on the calculation of inverse Hessian-vector products (iHVP), but the classical solver "Linear time Stochastic Second-order Algorithm" (LiSSA, Agarwal et al. (2017)) is often deemed impractical for large models due to expensive computation and hyperparameter tuning. We show that the three hyperparameters — the scaling factor, the batch size, and the number of steps — can be chosen depending on two specific spectral properties of the Hessian: its trace and largest eigenvalue. By evaluating them with random sketching (Swartworth and Woodruff, 2023), we find that the batch size has to be sufficiently large for the LiSSA to converge; however, for all of the models we consider, the requirement is mild. We confirm our findings empirically by comparing to the Proximal Bregman Retraining Functions (PBRF, Bae et al. (2022)).

## 1 INTRODUCTION

Deep neural networks have seen many impressive results in the past years, but researchers and practitioners have little understanding what happens inside the models and how they learn to predict. While there exist various methods to interpret the internal computations of neural networks in an understandable to a human way, influence functions attempt to explain model behaviour by attributing model predictions (or generations) to particular examples in the training data.

Koh and Liang (2017) introduce Hessian-based influence functions in order to approximate the effect of removal of one training point from the training set, which we refer to as leave-one-out retraining. The formula for influence calculation is derived from the second-order Taylor approximation of the loss, hence the Hessian and the gradient of the training point are sufficient for calculation. Koh and Liang (2017) demonstrate various applications of influence functions such as explaining of model outputs through data attribution, repairing mislabeled data, and backdoor attacks.

Basu et al. (2020) criticize influence functions for poor approximation of leave-one-out retraining as the depth and width of neural networks increase. As a solution, Bae et al. (2022) propose two fixes: replacing the Hessian (that possibly has negative eigenvalues) with the well-behaved Gauss-Newton Hessian (Martens, 2020), and replacing the leave-one-out retraining (which itself is not a well-defined objective) with the Proximal Bregman Retraining Functions (PBRF). They demonstrate that the latter do not suffer from the randomness introduced by model initialization and data sampling, and they argue can serve as gold standard when evaluating influence function approximation methods. In this paper, we focus on this particular formulation of influence functions, where the Hessian is replaced with Gauss-Newton Hessian, and the PBRF serves as the ground truth in validation.

The calculation of influence functions, as introduced by Koh and Liang (2017), requires approximation of inverse Hessian-vector products. Given the dimension of modern deep models and the size of the training dataset, it can be a hard problem. As an alternative to traditional conjugate gradient method, Koh and Liang (2017) proposed to use a stochastic iterative approach called "Linear time Stochastic Second-Order Algorithm" (LiSSA, (Agarwal et al., 2017)). This algorithm requires calculating a sampled Hessian-vector product at each iteration, where in the formulation of Koh and Liang (2017), the batch size per sample can be as little as just one training point. There are two additional hyperparameters involved — the scaling factor and the number of steps in the LiSSA, with little direction of how to choose them in practice. Basu et al. (2020) criticize the LiSSA, suggesting that it becomes erroneous for deep networks with large number of parameters, and they

also mention the importance of hyperparameter search. In this paper, we find that the choice of all three hyperparameters, including the batch size, depends on the properties of the Gauss-Newton Hessian, namely, its trace and largest eigenvalue. Since the size of the Hessian is very large (number of parameters to square), we evaluate these two statistics with *random sketching* (Swartworth and Woodruff, 2023), which only requires estimation of Hessian-vector products in the process. We report these statistics and the corresponding requirements for some open sourced vision and language models. We also note that, contrary to a common belief, we find that the batch size has to be sufficiently large for the algorithm to converge. However, this requirement is mild, and for language models it is particularly redundant.

Some attempts to avoid calculating inverse Hessian vector products have been made in the literature. Schioppa et al. (2022) suggest to truncate the spectrum of the Hessian, see also Fisher et al. (2023); Grosse et al. (2023). When it comes to language models, most of the recent literature is using gradient-based influence functions (Xia et al., 2024; He et al., 2024; Chhabra et al., 2024). These are typically focused on the finetunning stage, and often this choice is motivated by simpler and faster implementation. In Section 5, we highlight the difference between the results of these two ways of calculating influence functions.

Almost exclusively, Grosse et al. (2023) calculate Hessian-based influence of pretraining data for LLMs. Their analysis is restricted to the MLPs of the transformer and they impose a block-wise structure onto the Hessian. Although we do not advocate against such structural assumptions, our work suggests that running the plain and model-agnostic LiSSA can be feasible, given that we avoid the hyperparameter search. In our implementation, we follow the Hessian-free approach of Martens et al. (2010) using finite differences, where an in-batch GNH-vector product is calculated with three forward propagations and one backward propagation. For example, our code allows to run the LiSSA for 7B language models on a $4{\times}$A100 GPU node. We share the code here `https://anonymous.4open.science/r/gnhtools-53AE`.

## 2 BACKGROUND AND NOTATION

Influence functions (Koh and Liang, 2017) are calculated under the assumption that $\theta$ delivers minimum to the training loss,

$$\theta^* = \arg\min_\theta \frac{1}{|\mathcal{D}_{tr}|} \sum_{(\mathbf{x},y)\in\mathcal{D}_{tr}} \ell(\mathbf{x},y;\theta), \tag{1}$$

where for classification tasks, $(\mathbf{x}, y)$ are input and label pair, and for language modeling tasks, consists of context and next word token. That is, given a sequence $s = (s_1, \ldots, s_l)$, the dataset $\mathcal{D}_{tr}$ consists of pairs $\mathbf{x} = (s_1, \ldots, s_{t-1})$ and $y = s_t$. Let us fix a point $(\mathbf{x}_m, y_m) \in \mathcal{D}_{tr}$, and for a small perturbation weight $\epsilon > 0$ consider

$$\theta^*(\epsilon) = \arg\min_\theta \frac{1}{|\mathcal{D}_{tr}|} \sum_{(\mathbf{x},y)\in\mathcal{D}_{tr}} \ell(\mathbf{x},y;\theta) + \epsilon\,\ell(\mathbf{x}_m, y_m; \theta). \tag{2}$$

Then, the *influence of a training point* $(\mathbf{x}_m, y_m)$ on the parameter is denoted as

$$
\begin{aligned}
\mathcal{I}((\mathbf{x}_m, y_m)) &= \frac{d\theta^*(\epsilon)}{d\epsilon}\Big|_{\epsilon=0} \\
&= -H^{-1}\nabla\ell(\mathbf{x}_m, y_m; \theta^*),
\end{aligned}
\tag{3}
$$

where $H$ denotes the *population Hessian*, that is

$$H = \frac{1}{|\mathcal{D}_{tr}|} \sum_{(\mathbf{x},y)\in\mathcal{D}_{tr}} \nabla^2\ell(\mathbf{x},y;\theta^*).$$

Furthermore, suppose we have a set of predictions $z_{test} = (\mathbf{x}_{test}, \hat{y}_{test})$, and let $f((\mathbf{x}, y), \theta) = \log p(y|\mathbf{x}; \theta)$ are the log probability according to the trained model. Then, the influence of a *training point* $z_{train} = (\mathbf{x}_m, y_m)$ *on a prediction* $z_{test}$ is denoted as

$$\mathcal{I}(z_{train}, z_{test}) = -\nabla f(z_{test}, \theta)^\top H^{-1} \nabla\ell(\mathbf{x}_m, y_m; \theta^*)$$

For language models, we calculate the influence for the completion task following Grosse et al. (2023). Let $s = (s_1, \ldots, s_p)$ be a prompt and $\hat{s} = (\hat{s}_1, \ldots, \hat{s}_c)$ be a completion. Then, we calculate the influence for average log-probability of predicted tokens

$$f(s, \hat{s}; \theta) = \frac{1}{c} \sum_{j=1}^{c} \log p(\hat{s}_j | s_1 \ldots s_p \hat{s}_1 \ldots \hat{s}_{j-1}; \theta).$$

**The PBRF as ground truth.** The inverse problem $H^{-1} \nabla \ell(z_{train}; \theta)$ can be difficult to perform due to degenerate eigenvalues of $H$. Koh and Liang (2017) propose to use a damping parameter $\lambda > 0$ and instead invert a regularized matrix $(H + \lambda I)^{-1}$. However, such matrix can still be degenerate due to possibly negative eigenvalues of the Hessian of a non-convex loss, which are indeed observed in practice and are not necessarily small (Sagun et al., 2017; Schioppa, 2024). Motivated by natural descent methods, Bae et al. (2022) propose to replace it with *Gauss-Newton Hessian* (GNH), which is denoted as follows. Suppose that the loss has the form

$$\ell((\mathbf{x}, y); \theta) = \ell(h(\mathbf{x}; \theta), y), \qquad \ell(h, y) = -\log(\mathrm{sf}(h)_y),$$

where $h(\mathbf{x}; \theta) \in \mathbb{R}^K$ is the logit function and $\mathrm{sf}(h)_j = \exp(h_j) / \left( \sum_{j=1}^{K} \exp(h_j) \right)$ is the standard softmax function. Then, the GNH has the form

$$H = \frac{1}{|\mathcal{D}_{tr}|} \sum_{(\mathbf{x}, y) \in \mathcal{D}_{tr}} [J_\theta h(\mathbf{x}; \theta)] \nabla_h^2 \ell(h(\mathbf{x}; \theta), y) [J_\theta h(\mathbf{x}; \theta)]^\top \tag{4}$$

For the Cross-Entropy loss, we have the identity $\nabla_h^2 \ell(h(\mathbf{x}; \theta), y) = \mathrm{Diag}(\mathrm{sf}(h)) - \mathrm{sf}(h)\mathrm{sf}(h)^\top$.

Furthermore, Bae et al. (2022) show that if the Hessian is replaced with the Gauss-Newton Hessian, the influence functions (3) approximate a different type of retraining called the Proximal Bregman Retraining Functions (PBRF). These correspond to retraining of the Proximal Bregman Objective (PBO) on training point $(\mathbf{x}_m, y_m)$ reads as follows,

$$\theta_{PBRF}(\epsilon) = \arg \min_\theta \frac{1}{|\mathcal{D}_{tr}|} \sum_{(\mathbf{x}, y) \in \mathcal{D}_{tr}} D(h(\mathbf{x}; \theta), h(\mathbf{x}; \theta^\star), y) + \epsilon \ell((\mathbf{x}_m, y_m); \theta) + \frac{\lambda}{2} \|\theta - \theta^\star\|^2, \tag{5}$$

where $D(h, h', y) = \ell(h, y) - \ell(h', y) - (h - h')^\top \nabla_h \ell(h', y)$ is the Bregman divergence. Comparing the PBO with the objective in (2), we see that the proximity penalty $\frac{\lambda}{2} \|\theta - \theta^\star\|^2$ takes into account the damping parameter, while replacing the loss with the Bregman divergence accounts for potential lack of convergence, i.e. we no longer need to assume that the training of the original parameter converges to global minimum of the loss as in (1). In addition, Bae et al. (2022) find that the PBRFs are a more reliable objective compared to traditional retraining, which is known to produce inconsistent outputs depending on initialization (Basu et al., 2020).

Bae et al. (2022) argue that the PBRF is a suitable ground truth objective for validation of influence function estimation algorithms. For instance, it is used for empirical validation of the Kronecker factored approximation in Grosse et al. (2023). Following them, we refer to the PBRF as a ground truth influence in order to confirm our findings empirically.

**Iterative inverse Hessian-vector products.** For calculating these inverse Hessian-vector products (iHVP) of form $\mathbf{u} = (H + \lambda)^{-1}\mathbf{g}$, Koh and Liang (2017) propose to use a variant of Linear time Stochastic Second-Order Algorithm (LiSSA, Agarwal et al. (2017)), that consist of the iterations

$$\mathbf{u}^t = \mathbf{g} + (I - \eta(\tilde{H}^t + \lambda))\mathbf{u}^{t-1}, \qquad t = 1, \ldots, T \tag{6}$$

where $\tilde{H}^t$ is an in-batch estimate of $H$. Ideally, the scaling parameter $\eta > 0$ needs to be chosen to ensure that $\eta H$ is a contraction, however, it requires knowning the larges eigenvalue of $H$. To this day, the LiSSA is often discarded due to it's hyperparameters, which are not trivial to tune when one does not have a clear objective Basu et al. (2020). In particular, the three hyperparameters — $\eta$, $T$, and the batch size, are often chosen without any directive, and the resulting estimate is deemed unreliable Basu et al. (2020).

The LiSSA updates (6) are equivalent to the stochastic gradient descent (SGD) with step size $\eta$ for the quadratic objective $\min_\mathbf{u} \frac{1}{2}\mathbf{u}^\top(H + \lambda)\mathbf{u} - \mathbf{u}^\top\mathbf{g}$ (Fisher et al., 2023). In practice, an in-batch function

is usually not as smooth as the average over the whole dataset (Tang et al., 2020). Furthermore, the optimal choice of the step size $\eta$ and number of steps $T$ depends on the largest eigenvalue of the Hessian $\lambda_{\max}(H)$. Rather than conducting hyperparameter grid search, we suggest to evaluate the largest eigenvalue $\lambda_{\max}(H)$ directly, so that we only run the LiSSA once.

**Hessian-vector products.** The updates (6) involve calculation of in-batch Hessian-vector products $\tilde{H}_t\mathbf{u}$. Expanding the expression for GNH (4), we observe that for a batch of data $B = \{(\mathbf{x}, y)\}$, we have

$$\tilde{H}_t\mathbf{u} = \frac{1}{|B|} \sum_{(\mathbf{x},y)\in B} [J_\theta h(\mathbf{x};\theta)]\nabla_h^2 \ell(h(\mathbf{x};\theta), y)[J_\theta h(\mathbf{x};\theta)]^\top \mathbf{u} \tag{7}$$

Here $[J_\theta h(\mathbf{x};\theta)]^\top \mathbf{u}$ is a directional derivative of vector-function $h(\mathbf{x};\theta)$. Calculating the directional derivatives per each example in the batch precisely may be prohibitively expensive. Instead, we suggest to approximate it by finite differences,

$$[J_\theta h(\mathbf{x};\theta)]^\top \mathbf{u} = \frac{d}{d\delta} h(\mathbf{x};\theta + \delta\mathbf{u})\Big|_{\delta=0} \approx \frac{h(\mathbf{x};\theta + \delta\mathbf{u}) - h(\mathbf{x};\theta - \delta\mathbf{u})}{2\delta},$$

where $\delta$ is a small value, which we fix to $\delta = 0.01$ in our experiments. Then, we can approximate the in-batch GNH-vector prodcut by using three forward propagations and one backward propagation:

$$\tilde{H}_t\mathbf{u} \approx \nabla_\theta \frac{1}{|B|} \sum_{(\mathbf{x},y)\in B} h(\mathbf{x};\theta)^\top S_h \left\{ \frac{h(\mathbf{x};\dot\theta + \delta\mathbf{u}) - h(\mathbf{x};\dot\theta - \delta\mathbf{u})}{2\delta} \right\},$$

where $\dot\theta$ indicates that we do not calculate the derivative through this parameter, and $S_h = \mathrm{Diag}(\mathrm{sf}(h)) - \mathrm{sf}(h)\mathrm{sf}(h)^\top$ is also fixed. Thus, we simply backpropogate through a weighted sum of logits in the batch $h(\mathbf{x};\theta)$, with weights depending on the matrices $S_h$ and finite differences $(h(\mathbf{x};\dot\theta + \delta\mathbf{u}) - h(\mathbf{x};\dot\theta - \delta\mathbf{u}))/(2\delta)$. The latter two can be calculated in a gradient free manner with three forward propagations. We note that incorporating finite differences for Hessian-vector products calculation has previously been done in Martens et al. (2010); Martens and Sutskever (2011).

## 3 APPROXIMATION ERROR AND CHOICE OF HYPERPARAMETERS

In order to carefully analyze approximation error of LiSSA iterations (6) we reformulate them as SGD updates. Observe that the result of iHVP applied to a gradient $\mathbf{g}$, $\mathbf{u}^\star = (H + \lambda)^{-1}\mathbf{g}$, delivers minimum to the following objective

$$\mathbf{u}^\star = \arg\min_{\mathbf{u}} L(\mathbf{u}), \qquad L(\mathbf{u}) := \frac{1}{2}\mathbf{u}^\top H\mathbf{u} + \frac{\lambda}{2}\|\mathbf{u}\|^2 - \mathbf{u}^\top \mathbf{g}. \tag{8}$$

With appropriate scaling, the LiSSA updates are equivalent to SGD with step size $\eta$ and the gradient calculated on an in-batch loss $\tilde{L}_t(\mathbf{u})$, where the Gauss-Newton Hessian $H$ is replaced with unbiased estimate $\tilde{H}_t$ calculated over a random batch, turning the updates in (6) into

$$\mathbf{u}^t = \mathbf{u}^{t-1} - \eta\left[(\tilde{H}^t + \lambda)\mathbf{u}^{t-1} - \mathbf{g}\right], \qquad t = 1, \ldots, T$$

where the scaling parameter in (6) now plays the role of a learning rate. SGD is well studied in the literature, with the recommending step size $\eta$ typically depending on the smoothness of $L(\mathbf{u})$, which in our case equals to $\lambda_{\max}(H)$ (Bubeck et al., 2015). In theory, the literature typically focuses on studying mini-batch SGD Hardt et al. (2016). However, mini-batch SGD may perform poorly in practice due to the difference in smoothness of the population objective $L(\mathbf{u})$ and the in-batch objective $\tilde{L}(\mathbf{u})$ as pointed out by Tang et al. (2020). In particular, this difference is affected by the choice of batch size for evaluating the Hessian $\tilde{H}_t$ at every step. In the context of LiSSA, the original work Koh and Liang (2017) only mentions that the average $\mathbf{E}\mathbf{u}^t$ converges as long as $\eta < 1/(\lambda_{\max}(H) + \lambda)$. The subsequent work does not address the choice of batch size either (Basu et al., 2020; Bae et al., 2022).

In order to take the sampling error into account, we need to control the second moments of the HVPs $\tilde{H}_t u$. Although the behavior of the matrix $\mathbf{E}\tilde{H}_t^2$ can generally be arbitrary, we find that in many cases we can rely on the following simple condition

$$\mathbf{E}\tilde{H}_t^2 - H^2 \preceq \frac{C}{|B|}\mathrm{Tr}(H)H, \tag{C.1}$$

where $|B|$ is the batch size, which for language models corresponds to the total in-batch *number of tokens*. For simplicity, we assume this number to be the same in each batch $B$. We think of $C$ as a moderately large constant, e.g. $C = 2$.

Below we confirm that Condition C.1 holds under some simplified assumptions on the distribution of the gradients. We consider two cases. In the **classification case**, our training set $\mathcal{D}_{tr} = \{(\mathbf{x}, y)\}$ consists of pairs of inputs and labels. The batches used to sample the HVPs in (7) consist of independently drawn instances $\mathbf{x} \sim \mathcal{U}(\mathcal{D}_{tr})$. In the **language modeling case**, we sample the batches per sequence $s$. For simplicity, we assume that all sequences have the same length, $|s| = L + 1$, and if $s \in B$, and $|s| = L + 1$, our batch contains all pairs $(\mathbf{x}, y)$ with context $\mathbf{x} = (s_0, \ldots, s_{t-1})$ and label $y = s_t$ for $t = 1, \ldots, L$. Our training dataset consists of all such pairs $\mathcal{D}_{tr} = \{(\mathbf{x}, y)\}$.

In both cases, we consider the gradient $\mathbf{g} \stackrel{d}{=} \nabla\ell(\hat{y}|\mathbf{x})$, where $\mathbf{x} \sim \mathcal{U}(\mathcal{D}_{tr})$ and $\hat{y}|\mathbf{x} \sim p(\mathbf{x})$ is drawn according to the trained model predictions. We say that $\mathbf{g}$ satisfies a *bounded kurtosis* condition, if for every direction $u \in \mathbb{R}^{|\theta|}$,

$$\mathbf{E}^{1/4}(\mathbf{g}^\top u)^4 \leq \beta\mathbf{E}^{1/2}(\mathbf{g}^\top u)^2, \tag{9}$$

which is a popular assumption when dealing with covariances of heavy tailed distribution  This condition is sufficient to show C.1 in the independent sampling case. In the case of **language modeling**, we additionally account for dependencies between the tokens in the batch in the following way. We assume that there is a constant $R > 0$ such that for any sequence $s \in \mathcal{D}_{tr}$ and any two token $\mathbf{x} \in s$ it holds

$$\sum_{\mathbf{x}' \in s} \max_{y,y'} |\cos(\nabla\ell(\mathbf{x}, y), \nabla\ell(\mathbf{x}', y'))| \leq R, \tag{10}$$

where we denote the cosine similarity $\cos(a, b) = a^\top b/(\|a\| \cdot \|b\|)$. This can be reasonable to expect due to the large dimension of the parameter, and we also note that a similar assumption appears in Tang et al. (2020) in the context of imaging inverse problems. We can also understand this condition in the following way: each token within a training sequence is related to a finite amount of local tokens and a finite amount of keywords within the sequence.

**Lemma 1.** *Consider the gradients* $\mathbf{g} = \nabla\ell(\hat{y}|\mathbf{x}), \mathbf{x} \sim \mathcal{D}_{tr}, \hat{y}|\mathbf{x} \sim p(\mathbf{x})$. *Suppose the gradient* $\mathbf{g}$ *has bounded* kurtosis *in all directions, that is for some* $\beta > 1$ *and for any* $u \in \mathbb{R}^{|\theta|}$, $\left(\mathbf{E}(u^\top\mathbf{g})^4\right)^{1/4} \leq \beta\left(\mathbf{E}(u^\top\mathbf{g})^2\right)^{1/2}$. *Suppose that we either have a* **i) classification task**, *where instances in a batch is drawn independently, or* **ii) language modeling task**, *where the batch consist of* $k$ *independently drawn sequences. We assume that each sequence has the batch size* $L + 1$, *and the batch consists of total* $|B| = Lk$. *We additionally assume that (10) holds for some* $R \geq 0$. *Then, condition C.1 holds with* $C = C(\beta, R)$.

For proof, see Section B.2 in the appendix. In addition, we provide a simple empirical test where we compare the traces of LHS and RHS in (C.1), and it confirms the inverse scaling with batch size for both classification and language modeling tasks. We also evaluate that a low dimensional projection of the difference of RHS and LHS is positive semi-definite for a small GPT2 model. See Section C in the appendix.

Under condition C.1, we show the following bound with an exact requirement for a sufficiently large batch size.

**Theorem 1.** *Suppose that C.1 holds. Let us choose the hyperparameters*

$$\eta = 1/(\lambda_{\max}(H) + \lambda), \qquad |B| \geq C\mathrm{Tr}(H)/\lambda_{\max}(H)$$

*Then,*

$$\mathbf{E}\|\mathbf{u}^t - \mathbf{u}^\star\|^2 \lesssim (1 - \lambda\eta)^{2t}(\|\mathbf{u}^0\|^2 + \|\mathbf{u}^\star\|^2) + \frac{\eta^2\mathrm{Tr}(H)}{|B|}\mathbf{g}^\top(H + \lambda)^{-1}\mathbf{g}. \tag{11}$$

In our error bound above, the first term depends on the learning rate and the number of steps, and we can say that it measures how quickly we converged to the solution, therefore we label it *convergence*

Table 1: Gauss-Newton-Hessian statistics and recommended hyperparameters. Statistics are calculated on ImageNet (IN) on vision and Open-Web-Text-2 (OWT) on language models. The arrow $\uparrow$ indicates a lower bound, and $\downarrow$ indicates that it is an upper bound.

| Model | Size | Data | $\frac{1}{N}\mathrm{Tr}(H)$ | $\lambda_{\max}(H)$ from sketching | recommended hyperparameters$^*$ | | |
| | | | | | $\downarrow \eta$ | $\uparrow |B|$ | $\uparrow T$ |
|---|---|---|---|---|---|---|---|
| ResNet-18 | 11M | IN | $(1.32 \pm 0.00) \times 10^{-3}$ | $\approx 270$ | 0.003 | 100 | 150 |
| ResNet-50 | 25M | IN | $(8.17 \pm 0.11) \times 10^{-4}$ | $\approx 470$ | 0.002 | 5 | 200 |
| OPT | 1.3B | OWT | $(9.28 \pm 0.35) \times 10^{-6}$ | $\approx 780$ | 0.001 | 30 | 500 |
| Llama-1 | 7B | OWT | $(5.69 \pm 0.67) \times 10^{-6}$ | $\approx 1600$ | 0.0005 | 50 | 1000 |
| Mistral | 7B | OWT | $(8.18 \pm 0.13) \times 10^{-5}$ | $\approx 5600$ | 0.0002 | 200 | 2000 |

$^*$ Assuming $C = 2$ in C.1, and we take $\lambda = 5.0$, $T = 2/(\lambda\eta)$ for the sake of demonstration.

*error*. The second term depends directly on the batch size and we label it *sampling error*. It does not depend on the number of steps performed and comes from the difference between the sampled and population HVPs. In particular, it is trivial to see that it corresponds to the variance of a single update, $\mathbf{E}\|\mathbf{u}^t - \mathbf{E}[\mathbf{u}^t|\mathbf{u}^{t-1}]\|^2$, in the limit $\mathbf{u}^t \to \mathbf{u}^\star$. Notice that although the *convergence error* does not depend on the batch size explicitly, we have to satisfy the condition $|B| \geq C\mathrm{Tr}(H)/\lambda_{\max}(H)$ in order to reduce the error of approximation. Furthermore, based on the convergence error, we suggest to take $T = \Omega(1/(\eta\lambda))$ steps. In the experiments, we take $T = 2/(\lambda\eta)$. In Appendix C.1, we construct a counter-example demonstrating that for any potential Hessian $H$, there exists a classification task with a data distribution that satisfies all conditions of Theorem 1. In this setting, while the LiSSA algorithm exhibits mean convergence ($\mathbf{E}[\mathbf{u}^t - \mathbf{u}^\star] \to 0$), it fails to converge pathwise ($\mathbf{E}\|\mathbf{u}^t - \mathbf{u}^\star\|^2 \to \infty$) when the batch requirement is broken. In our counter-example, condition (C.1) holds with $C = 1$. In the next section, we also empirically demonstrate that convergence can slow down when the batch size is not sufficiently large, even if we average over proportional number of trials.

We also note that in terms of batch size requirements, our result conforms with Tang et al. (2020). However, they only make a qualitative characterization that mini-batch SGD is applicable in problems where $H$ has a fast decaying eigenspectrum. We also mention (Dieuleveut et al., 2017) who take into account the difference between in-sample and global objective smoothness for general optimization problems. However, their analysis requires $\eta \leq \mathrm{Tr}(H)$. The closest to our work is Ma et al. (2018), our proof closely follows their analysis of the linear regression case (Theorem 2). We note that they do not make clear connection to the trace of the Hessian in determining critical batch size. In addition, their lower bound only covers the isotropic case $H = cI$.

**Empirical analysis of eigenvalue statistics.** It is apparent that to choose hyperparameters correctly, we need to evaluate the statistics $\lambda_{\max}(H)$ and $\mathrm{Tr}(H)$. Since there is no way to calculate the Hessian explicitly, we resort to random feature methods that only require evaluation of HVPs.

Evaluating the trace is straightforward. We generate a series of quadratic forms $(\mathbf{g}_i^\top \tilde{H}_i \mathbf{g}_i)_{i=1}^N$, which consists of calculating HVPs and dot products for Gaussian vectors $\mathbf{g} \sim \mathcal{N}(0, \frac{1}{N}I)$. Their mean estimates the trace $\mathbf{E}\mathbf{g}^\top \tilde{H}\mathbf{g} = \mathbf{E}\mathbf{g}^\top H\mathbf{g} = \frac{1}{N}\mathrm{Tr}(H)$. Due to independence of observations, we can also evaluate standard error of this estimator. For evaluating the largest eigenvalue we use *random sketching*. That is, we evaluate a matrix $\hat{H} = \Phi H \Phi^\top$, with $\Phi \in \mathbb{R}^{d \times N}$ randomly generated in a way such that $\Phi_{i,j} \sim \mathcal{N}(0, 1/d)$. It is known that such sketches can evaluate the top eigenvalues of the original matrix $\hat{\lambda}_i(H) = \lambda_i(\hat{H}) - \frac{1}{d}\mathrm{Tr}(\hat{H})$ (Swartworth and Woodruff, 2023), with error of estimation negligible for the top eigenvalue. See calculation details in the appendix, Section D.

We report the results of these evaluations in Table 1 for 2 ResNets and 3 open-sourced language models. We also show recommendations for the choice of hyperparameters based on Theorem 1. Notice that contrary to the original idea of SGD, in all cases the recommended batch size is larger than 1. However, recall that for language modeling, the batch size is the amount of tokens in a batch,

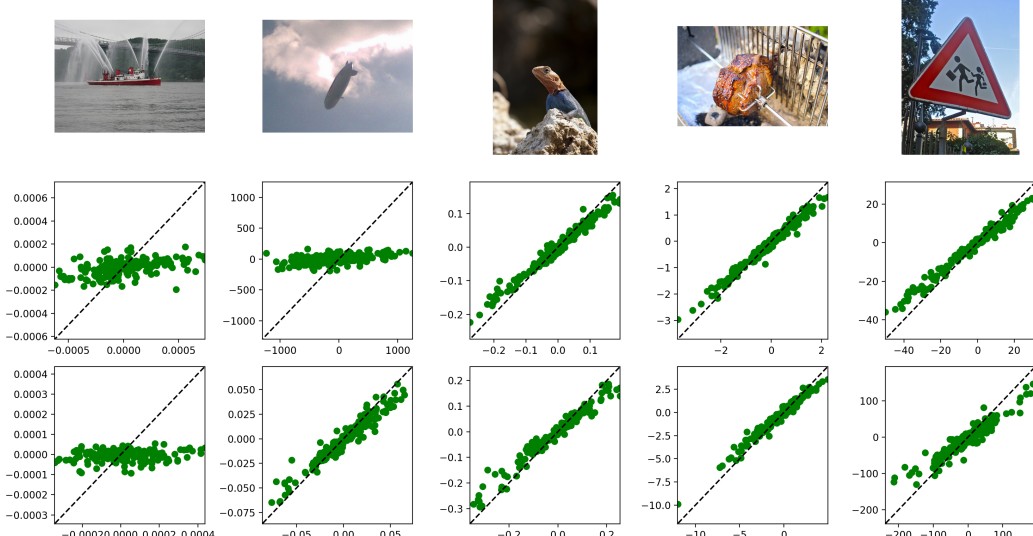

Figure 1: Comparison of the PBRF and the LiSSA influence. The first row shows examples of training images. Below, the $x$-axis represents LiSSA influences, and the $y$-axis represents the PBRF influences corresponding to each training image and 500 test images. The second row is for ResNet-18, and the third row is for ResNet-50.

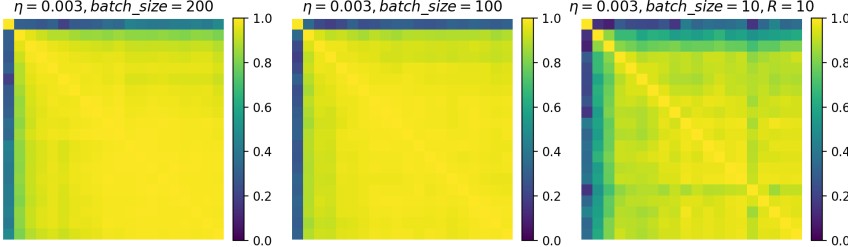

Figure 2: Convergence of LiSSA for ResNet-18 with different batch size configurations. We calculate the correlation between test influences at steps 1..1000 of LiSSA. The result for the small batch size of 10 is averaged over 10 trials, so that the amount of data used in the middle and the rightmost figures is the same.

and the recommended values are smaller than a typical context length. Thus, the LiSSA can work with just one sequence per batch. We also note that the recommendation is only a lower bound that ensures that LiSSA does not diverge, and increasing the batch size further makes the sampling error smaller (second term in (11)).

# 4 EMPIRICAL VALIDATION

We now conduct empirical validation of our theoretical results. In particular, we want to check two things. Firstly, we want to demonstrate that when the parameters are chosen according to Table 1, the LiSSA converges as expected. For ground truth, we calculate the PBRF for selected training examples Bae et al. (2022). Secondly, we want to empirically confirm that the requirement on sufficiently large batch size is indeed important.

We compare the LiSSA and the PBRF for the ResNet-18 and ResNet-50 models and randomly selected 25 training and 500 test images from the ImageNet dataset. For each training image we calculate the iHVP $s_{train} = (H + \lambda)^{-1}\nabla\ell(x_{train}, y_{train})$ using the LiSSA with hyperparameters from Table 1, and calculate the 500 influences $s_{train}^\top \nabla\ell(x_{test}, y_{test})$. We also calculate the the PBRF by finetuning the model with SGD on Proximal Bregman objective. For the PBRF, we match the batch size, number of steps, and the learning rate to the LiSSA. We take $\epsilon =$1e-8 in (5) and optimize the PBO using double precision to avoid float overflow. We show results for ResNet-18 and ResNet-50 for selected 5 images, and the full list is shown in the appendix, Section F. We observe three cases: 1) the LiSSA approximates the PBRF, i.e. scatter plot concentrates along the dashed line $x = y$; 2) both the LiSSA and PBRF have very low values poorly distinguishable from zero; 3) both the LiSSA and PBRF have high value and do not approximate each other. In the latter case, we can argue that the PBO finetunning stirs away the model too far for the quadratic approximation to hold.

Furthermore, we confirm that the batch size matters not only for the sampling error in (11), but also for the speed of convergence. Let us take ResNet-18 with damping parameter and run the LiSSA algorithm for 1000 steps. According to Table 1, the recommended batch size is equal to 100. We suggest to consider three set-ups: batch size 2x larger than recommended, batch size is equal to 100, and batch size 10x smaller than recommended, with the result averaged over 10 independent runs. In the latter and the former cases, the result is obtained through iterating in the same amount of data, which allows to equalize the sampling error of a single update. Figure 2 reports the correlation between influences calculated for 500 test images at different steps. As we can see, the correlation converges to 1 faster for the two cases where the batch size is greater or equal to the recommended size 100, despite averaging the iHVP over 10 trials for the smaller batch.

## 5 WHAT IS THE ROLE OF INVERTED HESSIAN?

In the context of language models, the focus in the current literature is mostly on gradient-based influence functions (Xia et al., 2024; He et al., 2024; Chhabra et al., 2024). Often this choice is motivated by simpler and faster implementation. Due to the high cost of Hessian-based influence calculation, it is natural to ask what are the benefits compared to the gradient-based influence. We conduct a simple experiment in an attempt to understand what is left out of the consideration when relying only on gradient dot products.

Consider the eigenvalue decomposition of the Gauss-Newton Hessian $H$,

$$H = \sum_{j=1}^{N} \lambda_j v_j v_j^\top,$$

where $v_j$ are orthogonal and normalized and $Hv_j = \lambda_j v_j$. If we represent a gradient $g = \nabla\ell(z_{test})$ in this eigenbasis, the iHVP simply reweights the coefficient according to how large the eigenvalue is,

$$g = \sum_j \langle g, v_j \rangle v_j, \qquad \lambda(H + \lambda)^{-1}g = \sum_j \frac{\lambda}{\lambda_j + \lambda} \langle g, v_j \rangle v_j. \tag{12}$$

Firstly, notice that as $\lambda \to \infty$, the iHVP in (12) converges to the plain gradient, while the price of the calculation decreases in accordance with Theorem 1. Therefore, the choice of the damping parameter offers a trade-off between the quality of approximation and computational complexity. Secondly, it is known that for classification tasks (what language models do per token), the Gauss-Newton Hessian is equivalent to a form of variance of the generated gradients $\nabla\ell(\hat{y}|x)$, where $\hat{y} \sim p(y|x)$, which is referred to as the Fisher Information Matrix (FIM). Generally speaking this is different from the empirical FIM $\mathbf{E}_{z_{train} \sim \mathcal{D}_{train}} \nabla\ell(z_{train})\nabla\ell(z_{train})^\top$, however, for realizable distributions the two might be used interchangeably (Kunstner et al., 2019). Such interpretation can help us to speculate, that the directions $v_j$ corresponding to higher eigenvalues $\lambda_j$ are more likely to observe in the training gradients, in the sense that $\mathbf{E}\langle g, v_j \rangle^2$ is higher. We also notice that $\frac{\lambda}{\lambda+\lambda_j} \approx 0$ for the top eigenvalues $\lambda_j$ that are much larger than the damping parameter $\lambda$. On the contrary, the lower eigenvalues receive higher weight, that is $\frac{\lambda}{\lambda+\lambda_j} \to 1$ as $\lambda_j \to 0$. In this sense, the iHVP works contrary to the traditional Principal Component Analysis, where the idea is to project the vector onto the top eigenvectors of the covariance. Instead, applying the inverse Hessian removes the top directions corresponding to $\lambda_j \gg \lambda$ and retains the directions corresponding to $\lambda_j \ll \lambda$.

Table 2: Examples of original and paraphrased sentences.

| original | *"The Mona Lisa, painted by Leonardo Da Vinci in the early 16th century, is one of the most famous paintings in the world."* |
|---|---|
| paraphrased | *"Leonardo Da Vinci's early 16th-century painting, the Mona Lisa, is widely regarded as one of the most renowned artworks globally."* |

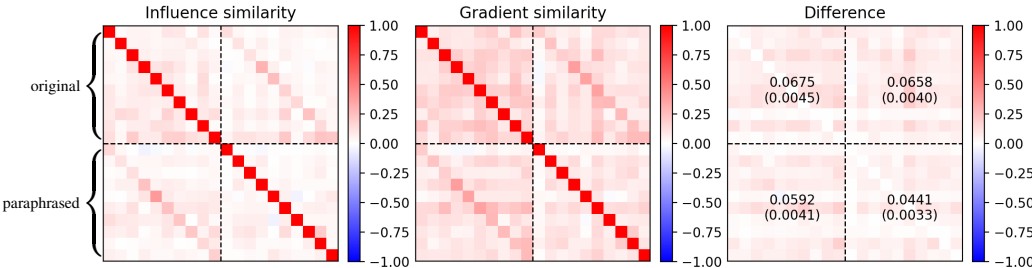

Figure 3: Similarity between 20 sentences, see complete list in in Appendix A. Left figure shows influence similarity calculated with the LiSSA, middle — gradient similarity, right — the difference between the former and the latter. In the rightmost figure the numbers show the mean over each 10x10 square, with standard error in the brackets. We use the OPT 1.3B model, with $\lambda = 5.0$, $T = 1000$ and $\eta = 0.003$. We also use batch size of 4 sequences, each consisting of 512 tokens.

For example, a plausible interpretation of the directions $v_j$ would be that the top directions correspond to general language coherence and sentence structure, while the remaining directions could correspond to more specific, informative content. We propose the following experiment to encourage such point of view. We consider ten pairs of sentences, one related to some historical or scientific fact, which we refer to as *original*, the other is a paraphrased version of the same fact, referred to as *paraphrased*. We show one such pair in Table 2, and in Appendix A we give all 10 pairs[1]. For every sentence, we calculate the gradient of the next word prediction loss $\nabla \ell(z)$ and calculate pairwise the dot-influences $\nabla \ell(z)^\top \nabla \ell(z')$ and the Hessian-based influences $\nabla \ell(z)^\top (H + \lambda)^{-1} \nabla \ell(z')$. Our goal is to measure the similarities between the original sentences and their rewritings. For this, we propose to measure the similarity by correspondingly normalizing with norms of gradients and self-influence:

$$\text{Gradient-similarity}(z, z') = \frac{\nabla \ell(z)^\top \nabla \ell(z')}{\|\nabla \ell(z)\| \|\nabla \ell(z')\|},$$

$$\text{Influence-similarity}(z, z') = \frac{\nabla \ell(z)^\top (H + \lambda)^{-1} \nabla \ell(z')}{\sqrt{\nabla \ell(z)^\top (H + \lambda)^{-1} \nabla \ell(z)} \sqrt{\nabla \ell(z')^\top (H + \lambda)^{-1} \nabla \ell(z')}}.$$

We show the pairwise similarities between all 20 sentences in Figure 3. In the rightmost graph, we also show the difference between the gradient similarity and the influence-based similarity. We observe that unrelated sentences generally have higher gradient similarity than influence similarity since the values in the rightmost graph are mostly positive. As a result, the influence similarity between an original sentence and a rewritten one appears to be consistently higher than between unrelated sentences.

Downweighting directions that are more likely to observe in (12) can also be compared to the idea of the TF-IDF index, where the terms are reweighted according to their inverse frequency (Salton and McGill, 1983). Incidentally, we show that for a bag-of-words model (which although trivial, is also a language model), the influence functions correspond to a particular form of the TF-IDF index, see Section E in the Appendix.

---

[1]To avoid cherry-picking, all 20 sentences were generated with Claude 3 Opus with a few prompts.

## 6 CONCLUSION

We have shown how to choose the hyperparameters of the classical LiSSA algorithm based on two spectral statistics of the Gauss-Newton Hessian. In particular, we show the batch size used for sampling Hessian-vector products per update has to be sufficiently large. Otherwise, the LiSSA might not converge which we demonstrate empirically and theoretically. This particular aspect of hyperparameter choice for the LiSSA algorithm has not been previously addressed in the literature. Furthermore, we empirically demonstrate that applying to large models in its original form can still be feasible if we choose a sufficiently large damping parameter. We do not necessarily advocate for using the LiSSA as the algorithm of choice, rather it can be used as a baseline for validating other more lightweight algorithms, in which case it is important to make sure that the hyperparameters of the LiSSA are chosen correctly. We hope that our result and the implementation can further facilitate research in influence functions, as well as in other topics where the inverse Hessian-vector products naturally appear (Guo et al., 2019; Schulman et al., 2015; Martens, 2020; Kirkpatrick et al., 2017).

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

## A  LIST OF PROMPTS FOR THE EXPERIMENT IN SECTION 5

1.

| | |
|---|---|
| original | *"The Great Wall of China is the longest wall in the world, stretching over 21,000 kilometers."* |
| rewrite | *"Spanning over 21,000 kilometers, the Great Wall of China holds the record for being the longest wall worldwide."* |

2.

| | |
|---|---|
| original | *"In 1969, Neil Armstrong became the first human to set foot on the Moon during the Apollo 11 mission."* |
| rewrite | *"During the Apollo 11 mission in 1969, Neil Armstrong made history by becoming the first person to walk on the lunar surface."* |

3.

| | |
|---|---|
| original | *"The theory of evolution by natural selection was first proposed by Charles Darwin in his book "On the Origin of Species" in 1859."* |
| rewrite | *"Charles Darwin introduced the concept of evolution by natural selection in his 1859 publication titled "On the Origin of Species"."* |

4.

| | |
|---|---|
| original | *"The United Nations was founded in 1945 after World War II to maintain international peace and security."* |
| rewrite | *"Following the conclusion of World War II, the United Nations was established in 1945 to foster international peace and security."* |

5.

| | |
|---|---|
| original | *"The Eiffel Tower, constructed in 1889 for the World's Fair, is one of the most iconic landmarks in Paris, France."* |
| rewrite | *"One of the most recognizable structures in Paris, France, the Eiffel Tower was built in 1889 for the World's Fair."* |

6.

| | |
|---|---|
| original | *"The French Revolution, which began in 1789, marked the end of the monarchy and the establishment of a republic in France."* |
| rewrite | *"The monarchy in France was abolished, and a republic was established as a result of the French Revolution, which commenced in 1789."* |

7.

| | |
|---|---|
| original | *"The human brain contains approximately 86 billion neurons, making it the most complex organ in the human body."* |
| rewrite | *"The most intricate organ in the human body, the brain, is composed of roughly 86 billion neurons."* |

8.

| | |
|---|---|
| original | *"In the 2020 United States presidential election, Joe Biden defeated incumbent Donald Trump to become the 46th president."* |
| rewrite | *"Joe Biden secured victory over the sitting president, Donald Trump, in the 2020 United States presidential election, becoming the 46th president."* |

9.

| | |
|---|---|
| original | *"The Mona Lisa, painted by Leonardo da Vinci in the early 16th century, is one of the most famous paintings in the world."* |
| rewrite | *"Leonardo da Vinci's early 16th-century painting, the Mona Lisa, is widely regarded as one of the most renowned artworks globally."* |

10.

| | |
|---|---|
| original | *"Climate change is a global issue caused by the increase of greenhouse gases in the atmosphere, primarily due to human activities."* |
| rewrite | *"The primary cause of climate change, a worldwide problem, is the accumulation of greenhouse gases in the atmosphere, largely attributed to human activities."* |

# B POSTPONED PROOFS FROM SECTION 3

## B.1 PROOF OF THEOREM 2

We first show a general convergence lemma for the updates (6), which repeats the steps of the proof of Theorem 2 in Ma et al. (2018).

**Lemma 2.** *Suppose, $\eta < 1/(\lambda_{\max}(H) + \lambda)$. Then, we have convergence in-expectation*

$$\|\mathbf{E}\mathbf{u}^t - \mathbf{u}^\star\| \le (1 - \lambda\eta)^t \|\mathbf{u}^0 - \mathbf{u}^\star\|.$$

*Furthermore, assume that $\eta > 0$, $\delta \in (0, 1)$ are such that*

$$(1 - \eta(H + \lambda))^2 + \eta^2(\mathbf{E}\tilde{H}_t^2 - H^2) \preceq (1 - \delta)I. \tag{13}$$

*Then,*

$$\mathbf{E}\|\mathbf{u}^t - \mathbf{u}^\star\|^2 \le (1 - \delta)^t \left(2\|\mathbf{u}^0 - \mathbf{u}^\star\|^2 + \|\mathbf{u}^\star\|^2\right) + \delta^{-1}\eta^2\tilde{\Delta},$$

*where we interpret $\tilde{\Delta} = \mathbf{E}\|(H - \tilde{H}_t)\mathbf{u}^\star\|^2$ as a sampling error.*

*Proof.* We write,

$$
\mathbf{u}^t - \mathbf{u}^\star = \left((1 - \lambda\eta)I + \eta\tilde{H}_t\right)\mathbf{u}^{t-1} + \eta\mathbf{g}
$$

$$
= \left(I - \eta(\tilde{H}_t + \lambda)\right)(\mathbf{u}^t - (H + \lambda)^{-1}\mathbf{g}) + \eta(H - \tilde{H}_t)(H + \lambda)^{-1}\mathbf{g}
$$

$$
= \left(I - \eta(\tilde{H}_t + \lambda)\right)(\mathbf{u}^{t-1} - \mathbf{u}^\star) + \eta(H - \tilde{H}_t)\mathbf{u}^\star .
$$

First, taking the expectation and using the fact that $\tilde{H}_t$ and $\mathbf{u}^t$ are independent,

$$
\mathbf{E}\mathbf{u}^t - \mathbf{u}^\star = (I - \eta(H + \lambda))(\mathbf{E}\mathbf{u}^{t-1} - \mathbf{u}^\star)
$$

$$
= (I - \eta(H + \lambda))^t(\mathbf{u}^0 - \mathbf{u}^\star),
$$

and under $\eta < 1/\lambda_{\max}(H)$ the matrix $I - \eta(H + \lambda) \preceq (1 - \lambda\eta)I$ is a contraction, thus the first bound follows.

For the second part, denote $R = \mathbf{E}(I - \eta(\tilde{H}_t + \lambda))^2$. Let us take the conditional expectation of the square norm, conditional on all the sampling before step $t$. Setting $\mathcal{F}_{t-1} = \sigma(\tilde{H}_1, \ldots, \tilde{H}_{t-1})$, we have that

$$
\mathbf{E}\left[\|\mathbf{u}^t - \mathbf{u}^\star\|^2|\mathcal{F}_{t-1}\right] = (\mathbf{u}^{t-1} - \mathbf{u}^\star)^\top \left[\mathbf{E}(I - \eta(\tilde{H}_t + \lambda))^2\right](\mathbf{u}^{t-1} - \mathbf{u}^\star)
$$

$$
+ 2\eta(\mathbf{u}^{t-1} - \mathbf{u}^\star)^\top\mathbf{E}(I - \eta(\tilde{H}_t + \lambda))(H - \tilde{H}^t)\mathbf{u}^\star
$$

$$
+ \eta^2\mathbf{E}\|(H - \tilde{H}_t)\mathbf{u}^\star\|^2
$$

$$
= \|R^{1/2}(\mathbf{u}^{t-1} - \mathbf{u}^\star)\|^2 + 2\eta^2(\mathbf{u}^{t-1} - \mathbf{u}^*)^\top\{\mathbf{E}\tilde{H}_t^2 - H^2\}\mathbf{u}^\star
$$

$$
+ \eta^2[\mathbf{u}^\star]^\top\{\mathbf{E}\tilde{H}_t^2 - H^2\}\mathbf{u}^\star
$$

$$
\leq (1 - \delta)\|\mathbf{u}^{t-1} - \mathbf{u}^\star\|^2 + 2\eta(\mathbf{u}^{t-1} - \mathbf{u}^\star)^\top\{\mathbf{E}\tilde{H}_t^2 - H^2\}\mathbf{u}^\star + \eta^2\tilde{\Delta} .
$$

Here, we have used the fact that by our assumption $\mathbf{E}(I - \eta(\tilde{H}_t + \lambda))^2 \preceq (1 - \delta)I$ is a contraction, since we have that

$$
\mathbf{E}(I - \eta(\tilde{H}_t + \lambda))^2 = (1 - \lambda\eta)^2I - 2\eta(1 - \lambda\eta)\mathbf{E}\tilde{H}_t + \eta^2\mathbf{E}\tilde{H}_t^2
$$

$$
= (1 - \lambda\eta)^2I - 2\eta(1 - \lambda\eta)H + \eta^2H^2
$$

$$
+ \eta^2(\mathbf{E}\tilde{H}_t^2 - H^2)
$$

$$
= (1 - \eta(H + \lambda))^2 + \eta^2(\mathbf{E}\tilde{H}_t^2 - H^2) .
$$

Taking the unconditional expectation, we obtain that

$$
\mathbf{E}\|\mathbf{u}^t - \mathbf{u}^\star\|^2 \leq (1 - \delta)\mathbf{E}\|\mathbf{u}^{t-1} - \mathbf{u}^\star\|^2 + 2\eta^2(\mathbf{u}^0 - \mathbf{u}^*)^\top(I - \eta(H + \lambda))^{t-1}\{\mathbf{E}\tilde{H}_t^2 - H\}\mathbf{u}^\star + \eta^2\tilde{\Delta}
$$

$$
\leq \ldots
$$

$$
\leq (1 - \delta)^t\|\mathbf{u}^0 - \mathbf{u}^\star\|^2 + \eta^2\tilde{\Delta}\left(1 + (1 - \delta) + \cdots + (1 - \delta)^{t-1}\right)
$$

$$
+ 2\eta^2(\mathbf{u}^0 - \mathbf{u}^\star)^\top\left\{\sum_{k=0}^{t-1}(1 - \delta)^k(I - \eta(H + \lambda))^{t-k}\right\}\{\mathbf{E}\tilde{H}_t^2 - H^2\}\mathbf{u}^\star
$$

$$
\leq (1 - \delta)^t\|\mathbf{u}^0 - \mathbf{u}^\star\|^2 + \frac{\eta^2\tilde{\Delta}}{1 - (1 - \delta)}
$$

$$
+ 2\eta^2(\mathbf{u}^0 - \mathbf{u}^\star)^\top\left\{(1 - \delta)^tI - (I - \eta(H + \lambda))^t\right\}(\eta(H + \lambda) - \delta I)^{-1}\{\mathbf{E}\tilde{H}_t^2 - H^2\}\mathbf{u}^\star .
$$

We apply the Cauchy-Schwartz inequality to the last term. By (13), we have that $\eta^2\{\mathbf{E}\tilde{H}_t^2 - H^2\} \preceq \eta(H + \lambda) - \delta I$. Thus,

$$
\left\|\left\{(1 - \delta)^tI - (I - \eta(H + \lambda))^t\right\}^{1/2}(\eta(H + \lambda) - \delta I)^{-1}\eta^2\{\mathbf{E}\tilde{H}_t^2 - H^2\}\mathbf{u}^\star\right\|^2
$$

$$
\leq (1 - \delta)^t\left\|(\eta(H + \lambda) - \delta I)^{-1}\eta^2\{\mathbf{E}\tilde{H}_t^2 - H^2\}\mathbf{u}^\star\right\|^2 \leq (1 - \delta)^t\|\mathbf{u}^\star\|^2 .
$$

We also have,

$$\left\| \left\{ (1-\delta)^t I - (I - \eta(H+\lambda))^t \right\}^{1/2} (\mathbf{u}^0 - \mathbf{u}^\star) \right\|^2 \le (1-\delta)^t \|\mathbf{u}^0 - \mathbf{u}^\star\|^2.$$

Collecting everything together,

$$\mathbf{E}\|\mathbf{u}^t - \mathbf{u}^\star\|^2 \le (1-\delta)^2(\|\mathbf{u} - \mathbf{u}^\star\|^2 + \|\mathbf{u} - \mathbf{u}^\star\|\|\mathbf{u}^\star\|) + \delta^{-1}\eta^2\tilde{\Delta}.$$

$\square$

Now we can complete the proof of Theorem 1.

We want to show that with $\frac{C\mathrm{Tr}(H)}{|B|} \ge \eta^{-1} - \lambda$, equation (13) takes place with $\delta = 2\eta\lambda - (\eta\lambda)^2$. Denote $K = \frac{C\mathrm{Tr}(H)}{|B|}$. Since Condition C.1 holds, we need to show

$$(I - \eta(\lambda + H))^2 + K\eta^2 H \preceq (1-\delta)I$$

The LHS of the above display has eigenvalues $(1-\lambda(\eta+\lambda_j))^2 + K\eta^2\lambda_j$, where $\lambda_j$ are the eigenvalues of $H$. It is therefore sufficient to show that

$$\max_{a\in[0,\lambda_{\max}]} (1 - \eta(\lambda + a))^2 + K\eta^2 a \le 1 - \delta$$

We rewrite this condition as

$$\max_{a\in[\lambda,\lambda+\lambda_{\max}]} -2\eta a + \eta^2 a^2 + K\eta^2(a - \lambda) \le -\delta$$

The minimum of the quadratic function is attained at $\bar{a} = \eta^{-1} - K/(2\eta)$. In the case where $\bar{a}$ is in the right half of the interval $[\lambda, \lambda + \lambda_{\max}]$, the maximum of the quadratic function is attained at the point $a = \lambda$. This condition rewrites as $\eta^{-1} - K/(2\eta) \ge \lambda + \lambda_{\max}/2$, which using $\eta(\lambda + \lambda_{\max}) = 1$ translates into $K \ge \eta^{-1} - \lambda$. Thus, under the assumption that the batch size is at least $|B| \ge \frac{C\mathrm{Tr}(H)}{\eta^{-1}-\lambda}$, we have that

$$\max_{a\in[0,\lambda_{\max}]} (1 - \eta(\lambda + a))^2 + K\eta^2 a = (1 - \eta\lambda)^2 \le 1 - \delta, \qquad \delta = 2\eta\lambda - (\eta\lambda)^2.$$

With such $\lambda$ it holds $(1 - \delta)^t = (1 - \eta\lambda)^{2t}$. It is left to notice that

$$\eta^2\|\mathbf{E}(\tilde{H}_t - H)\mathbf{u}^\star\|^2 = \eta^2[\mathbf{u}^\star]^\top \{\mathbf{E}\tilde{H}_t^2 - H^2\}\mathbf{u}^\star$$

$$\le \frac{(1+c)\mathrm{Tr}(H)}{|B|}\mathbf{g}^\top(H+\lambda)^{-1}H(H+\lambda)^{-1}\mathbf{g}$$

$$\le \frac{(1+c)\mathrm{Tr}(H)}{|B|}\mathbf{g}^\top(H+\lambda)^{-1}\mathbf{g}.$$

## B.2 PROOF OF LEMMA 1

**Proof for case i), classification task.** We first consider the case where the observations in each batch are drawn independently.

The in-batch Gauss-Newton Hessian then reads as

$$\tilde{H}_t = \frac{1}{|B|}\sum_{\mathbf{x}\in B}\mathbf{E}_{\hat{y}\sim p(\mathbf{x})}\nabla\ell(\hat{y}|\mathbf{x})\nabla\ell(\hat{y}|\mathbf{x})^\top$$

Thanks to the fact that the elements in $B$ are i.i.d. we have that

$$\mathbf{E}\tilde{H}_t^2 = \frac{1}{|B|}\mathbf{E}\sum_{\mathbf{x}\neq\mathbf{x}'}\mathbf{E}_{\hat{y}\sim p(\mathbf{x})}\nabla\ell(\hat{y}|\mathbf{x})\nabla\ell(\hat{y}|\mathbf{x})^\top\mathbf{E}_{\hat{y}\sim p(\mathbf{x}')}\nabla\ell(\hat{y}|\mathbf{x}')\nabla\ell(\hat{y}|\mathbf{x}')^\top$$

$$+ \frac{1}{|B|^2}\sum_{\mathbf{x}\in B}\left[\mathbf{E}_{\hat{y}\sim p(\mathbf{x})}\nabla\ell(\hat{y}|\mathbf{x})\nabla\ell(\hat{y}|\mathbf{x})^\top\right]^2$$

$$= (1 - 1/|B|)H^2 + \frac{1}{|B|^2}\sum_{\mathbf{x}\in B}\left[\mathbf{E}_{\hat{y}\sim p(\mathbf{x})}\nabla\ell(\hat{y}|\mathbf{x})\nabla\ell(\hat{y}|\mathbf{x})^\top\right]^2$$

$$\preceq (1 - 1/|B|)H^2 + \frac{1}{|B|}\mathbf{E}\|\nabla\ell(\hat{y}|\mathbf{x})\|^2\nabla\ell(\hat{y}|\mathbf{x})\nabla\ell(\hat{y}|\mathbf{x})^\top.$$

Let us write $\mathbf{g}$ instead of $\nabla\ell(\hat{y}|\mathbf{x})$. Observe that by the bounded kurtosis condition,

$$
\begin{aligned}
v^\top[\mathbf{E}\|\mathbf{g}\|^2\mathbf{g}\mathbf{g}^\top]v = \sum_j \mathbf{E}\langle\mathbf{g},e_j\rangle^2\langle v,\mathbf{g}\rangle^2 &\leq \sum_j \mathbf{E}^{1/2}\langle\mathbf{g},e_j\rangle^4\mathbf{E}^{1/2}\langle\mathbf{g},v\rangle^4 \\
&\leq \sum_j \beta^2\mathbf{E}\langle\mathbf{g},e_j\rangle^2\beta^2\mathbf{E}\langle\mathbf{g},v\rangle^2 \\
&= \beta^4\mathbf{E}\|\mathbf{g}\|^2\mathbf{E}\langle\mathbf{g},v\rangle^2 \\
&= \beta^4\mathrm{Tr}(H)(v^\top Hv),
\end{aligned}
$$

therefore, $\mathbf{E}\|\mathbf{g}\|^2\mathbf{g}\mathbf{g}^\top \preceq \beta^4\mathrm{Tr}(H)H$. Thus, C.1 holds with $C = \beta^4$.

**Proof for case ii), language modeling task.** Let us now consider the case where we sample the tokens sequence-wise. We assume that each sequence has the same size, so that each token in the dataset has an equal probability to be drawn. When we say a token is drawn, we mean that we consider the prediction of token $s_t$ with context $(s_0,\ldots,s_{t-1})$. This means that when $s \in B$, and $|s| = L+1$, our batch contains all pairs $(\mathbf{x},y)$ with contexts $\mathbf{x} = (s_0,\ldots,s_{t-1})$ and labels $y = s_t$ for $t = 1,\ldots,L$. In this case we also write $\mathbf{x} \in s$.

Let $B = s^{(1)} \cup \cdots \cup s^{(b)}$, where $b$ is the number of sequences in a batch, so that $|B| = bL$, and we assume that each sequence has the same length $L+1$ (we do not predict the first token in a sequence, whose index is 0). Let $\tilde{H}(s)$ denotes in-sequence GNH. Then,

$$
\mathbf{E}\tilde{H}_t^2 = H^2 + \frac{1}{b}\{\mathbf{E}\tilde{H}(s)^2 - H^2\}, \tag{14}
$$

where $s$ is a single random sequence. Let us expand,

$$
\tilde{H}(s)^2 = \frac{1}{L^2}\sum_{\mathbf{x},\mathbf{x}'\in s}\tilde{H}(\mathbf{x})\tilde{H}(\mathbf{x}')
$$

We have that for $\mathbf{x},\mathbf{x}' \in s$,

$$
\begin{aligned}
&\frac{1}{2}(\tilde{H}(\mathbf{x})\tilde{H}(\mathbf{x}') + \tilde{H}(\mathbf{x}')\tilde{H}(\mathbf{x})) \\
&= \frac{1}{2}\mathbf{E}_{\hat{y}\sim p(\mathbf{x}),\hat{y}'\sim p(\mathbf{x}')}\langle\nabla\ell(\hat{y}|\mathbf{x}),\nabla\ell(\hat{y}'|\mathbf{x}')\rangle(\nabla\ell(\hat{y}|\mathbf{x})\nabla\ell(\hat{y}'|\mathbf{x}')^\top + \nabla\ell(\hat{y}'|\mathbf{x}')\nabla\ell(\hat{y}|\mathbf{x})^\top) \\
&\preceq \frac{\delta(\mathbf{x},\mathbf{x}')}{2}\left(\mathbf{E}_{\hat{y}\sim p(\mathbf{x})}\|\nabla\ell(\hat{y}|\mathbf{x})\|^2\nabla\ell(\hat{y}|\mathbf{x})\nabla\ell(\hat{y}|\mathbf{x})^\top + \mathbf{E}_{\hat{y}'\sim p(\mathbf{x}')}\|\nabla\ell(\hat{y}'|\mathbf{x}')\|^2\nabla\ell(\hat{y}'|\mathbf{x}')\nabla\ell(\hat{y}'|\mathbf{x}')^\top\right),
\end{aligned}
$$

where we denote for short $\delta(\mathbf{x},\mathbf{x}') = \max_{\hat{y},\hat{y}'}|\cos(\nabla\ell(\hat{y},\mathbf{x}),\nabla\ell(\hat{y}',\mathbf{x}'))|$, and we also use the fact that $2\|a\|\|b\|\langle v,a\rangle\langle v,b\rangle \leq \|a\|^2\langle v,a\rangle^2 + \|b\|^2\langle v,b\rangle^2$. Summing up we have that

$$
\begin{aligned}
\frac{1}{L^2}\sum_{\mathbf{x},\mathbf{x}'\in s}\tilde{H}(\mathbf{x})\tilde{H}(\mathbf{x}') &\preceq \frac{1}{L^2}\sum_{\mathbf{x}\in s}\left(\sum_{\mathbf{x}'\in s}\delta(\mathbf{x},\mathbf{x}')\right)\mathbf{E}_{\hat{y}\sim p(\mathbf{x})}\|\nabla\ell(\hat{y}|\mathbf{x})\|^2\nabla\ell(\hat{y}|\mathbf{x})\nabla\ell(\hat{y}|\mathbf{x})^\top \\
&\preceq \frac{R}{L^2}\sum_{\mathbf{x}\in s}\mathbf{E}_{\hat{y}\sim p(\mathbf{x})}\|\nabla\ell(\hat{y}|\mathbf{x})\|^2\nabla\ell(\hat{y}|\mathbf{x})\nabla\ell(\hat{y}|\mathbf{x})^\top,
\end{aligned}
$$

where we used the fact that $\sum_{\mathbf{x}'\in s}\delta(\mathbf{x},\mathbf{x}') \leq R$ by the condition of the lemma. Now we take the expectation with respect to $s \sim \mathcal{U}(\mathcal{D}_{tr})$,

$$
\mathbf{E}\tilde{H}(s) \preceq \frac{R}{L}\mathbf{E}\|\nabla\ell(\hat{y}|\mathbf{x})\|^2\nabla\ell(\hat{y}|\mathbf{x})\nabla\ell(\hat{y}|\mathbf{x})^\top,
$$

where the latter expectation is with respect to the global sampling $\mathbf{x} \sim \mathcal{U}(\{\mathbf{x} \in s : s \in \mathcal{D}_{tr}\})$. The proof is completed following the corresponding steps of the independent sampling case, where we use the bounded kurtosis condition. We conclude that in this case the condition holds with $C = R\beta^4$.

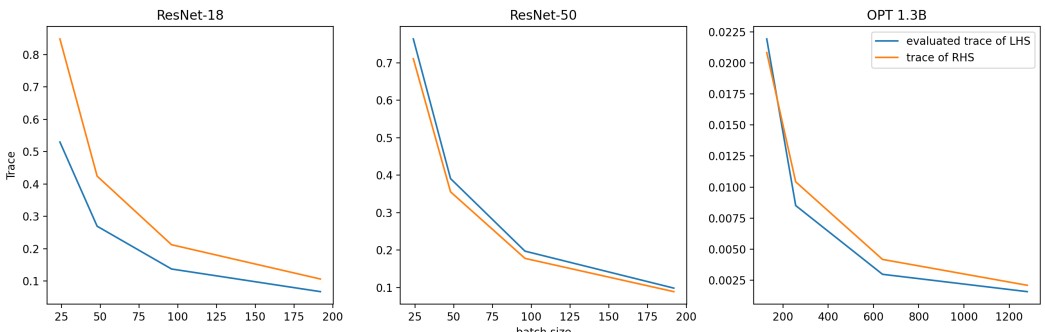

Figure 4: Compring traces of LHS and RHS of condition C.1 for different batch sizes. We evaluate the traces for ResNet-18, ResNet-50, and OPT-1.3B, 4 batch sizes for each model. For the OPT-1.3B, the batch size is counted in tokens.

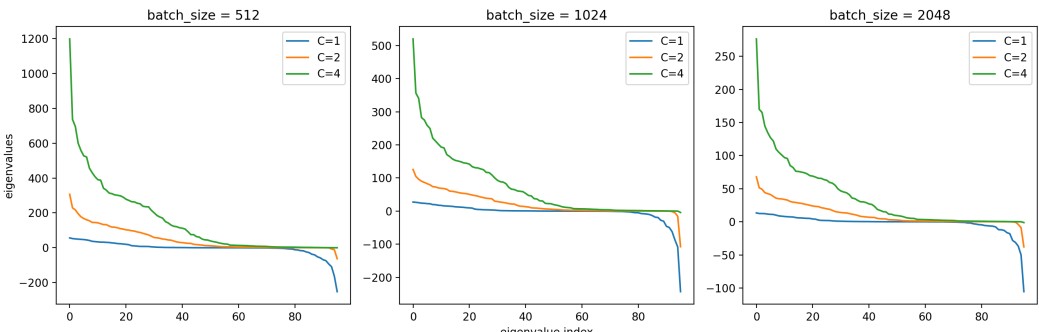

Figure 5: Spectrum of a random projection of difference RHS and LHS in the Condition C.1 with $C = 1, 2, 4$ and batch sizes $|B| = 512, 1024, 2048$. We evaluate the projections of dimension $d = 96$ for a small GPT2 model.

## C   EMPIRICAL CHECK OF CONDITION C.1

We propose a simple empirical sanity check by assessing the relationship between the LHS and RHS in condition (C.1). In Figure 4, we compare the traces of two matrices, similarly evaluating the traces by averaging over random quadratic forms $\mathbf{g}^\top (\tilde{H}_t^2 - H^2)\mathbf{g}$. We evaluate this gap by estimating the HVPs $\tilde{H}_t\mathbf{g}$, on random Gaussian vectors and taking their norm. Each evaluation is averaged over 1000 realizations of $\mathbf{g}$.

In addition to that, we evaluate the spectrum of random projections

$$\Phi^\top \left( \frac{C\,\mathrm{Tr}(H)}{|B|} H - \mathbf{E}\tilde{H}_t^2 + H^2 \right) \Phi,$$

where $\Phi = [\phi_1, \dots, \phi_d]$ consists of random Gaussian vectors as described in Section D. We note that evaluating $\mathbf{E}\tilde{H}_t^2\mathbf{g}$ requires applying HVP twice, which as we find is rather noisy. We average these over 1000 independent evaluations. To narrow down the cost of this experiment, we only apply it to the small GPT2 model and we take relatively small dimension $d = 96$. We show the spectrum of the matrix in the above display for $C = 1, 2, 4$, and $|B| = 512, 1024, 2048$ in Figure 5. We see that the spectrum is non-negative for $C = 4$.

### C.1   COUNTER-EXAMPLE WITH DIVERGENCE

**Lemma 3.** *There exists a binary classification task, where condition C.1 holds with exact equality*

$$\mathbf{E}\tilde{H}_t^2 - H^2 = \frac{1}{|B|}\mathrm{Tr}(H)H,$$

*and for some inputs* $\mathbf{u}^0, \mathbf{g}$ *the following claim holds.*

*Let us choose step size* $\eta = 1/(\lambda_{\max}(H) + \lambda)$, *and the batch size* $|B| < \mathrm{Tr}(H)/(\lambda_{\max}(H) + \lambda)$. *Then, the LiSSA algorithm converges on average, but not samplewise:*

$$\|\mathbf{E}\mathbf{u}^t - \mathbf{u}^\star\| \to 0, \qquad \mathbf{E}\|\mathbf{u}^t - \mathbf{u}^\star\|^2 \to \infty.$$

*Proof.* We consider a binary regression with $y \in \{0, 1\}$ and inputs $\mathbf{x} \in \mathbb{R}^N$. For simplicity, we assume that $\mathbf{x}, y$ are independent, and $y$ takes values $\{0, 1\}$ with equal probabilities. We consider the distribution in $\mathbf{x}$ that satisfies $\|\mathbf{x}\|^2 = \mathrm{Tr}(H)$ pointwise and $\mathbf{E}\mathbf{x}\mathbf{x}^\top = H$. For this, let $H = V\Lambda V^\top$ where $\Lambda = \mathrm{diag}\{\lambda_1, \ldots, \lambda_N\}$ and take $\mathbf{x} = V\mathbf{s}$, $\mathbf{s} = (\sqrt{\lambda_1}\epsilon_1, \ldots, \sqrt{\lambda_N}\epsilon_N)^\top$ where $\epsilon_i = \pm 1$ with equal probabilities. Consider the binary logistic model

$$\log p(y = 1|\mathbf{x}) = \theta^\top \mathbf{x} - \log(\exp(\theta^\top \mathbf{x}) + \exp(-\theta^\top \mathbf{x})),$$
$$\log p(y = 0|\mathbf{x}) = -\theta^\top \mathbf{x} - \log(\exp(\theta^\top \mathbf{x}) + \exp(-\theta^\top \mathbf{x})),$$

and assume that during training the model converged to the optimal parameters $\theta = 0$, since $\mathbf{x}, y$ are independent. Then it is straightforward to calculate that

$$\tilde{H}_t = \frac{1}{|B|} \sum_{(\mathbf{x},y) \in B} \mathbf{x}\mathbf{x}^\top, \qquad \mathbf{E}\tilde{H}_t^2 = \left(1 - \frac{1}{|B|}\right) H^2 + \frac{1}{|B|}\mathrm{Tr}(H)H.$$

Since all matrices are rotated by $V$ from left and right, we assume w.l.o.g. that $V = I$.

With such choice of $\eta$ and $|B|$, we have that the matrix $(1 - \eta(H + \lambda))^2 + \eta^2\{\mathbf{E}\tilde{H}_t^2 - H^2\}$ has eigenvalue strictly greater than 1, corresponding to the direction of the top eigenvalue of $H$. Let us denote this eigenvalue $\bar{\lambda}$.

Now assume that $\mathbf{g} = 0$ we have $\mathbf{u}^\star = 0$, and let $\mathbf{u}^0 \neq 0$. We have

$$\mathbf{u}^t = (1 - \eta(\tilde{H}_t + \lambda))\mathbf{u}^{t-1} = \prod_{j=1}^{t}(1 - \eta(\tilde{H}_j + \lambda))\mathbf{u}^0.$$

Set $Q = I - \eta(\tilde{H}_t + \lambda)$ and $R = \mathbf{E}(1 - \eta(\tilde{H}_t + \lambda))^2 = (1 - \eta(H + \lambda))^2 + \eta^2(\mathbf{E}\tilde{H}_t^2 - H^2)$. Notice that both matrices are diagonal. Consider the sequence of matrices $R_0 = I$, $R_1 = R$, $R_k = \mathbf{E}(1 - \eta(\tilde{H}_t + \lambda))R_{k-1}(1 - \eta(\tilde{H}_t + \lambda))$. Also denote partial product $\tilde{T}_j = \prod_{k=1}^{j}(1 - \eta(\tilde{H}_k + \lambda))$. Then we have

$$\begin{aligned}
\mathbf{E}\|\mathbf{u}^t\|^2 &= [\mathbf{u}^0]^\top \mathbf{E}C_{t-1}(1 - \eta(\tilde{H}_t + \lambda))^2 C_{t-1}\mathbf{u}^0 \\
&= [\mathbf{u}^0]^\top \mathbf{E}T_{t-1}RC_{t-1}\mathbf{u}^0 \\
&= [\mathbf{u}^0]^\top \mathbf{E}T_{t-2}(1 - \eta(\tilde{H}_{t-1} + \lambda))R_1(1 - \eta(\tilde{H}_{t-2} + \lambda))C_{t-2}\mathbf{u}^0 \\
&= [\mathbf{u}^0]^\top \mathbf{E}T_{t-2}R_2 T_{t-2}\mathbf{u}^0 \\
&= [\mathbf{u}^0]^\top \mathbf{E}T_{t-3}R_3 T_{t-3}\mathbf{u}^0 \\
&= \ldots \\
&= [\mathbf{u}^0]^\top R_t\mathbf{u}^0
\end{aligned}$$

Let us show that the matrix $B_t$ is diagonal. Indeed, assuming $B_{k-1}$ is diagonal, we have that

$$\begin{aligned}
R_k &= QR_{k-1}Q + \eta^2\mathbf{E}(\tilde{H}_t - H)R_{k-1}(\tilde{H}_t - H) \\
&= QB_{k-1}Q - \eta^2 HR_{k-1}H + \eta^2\mathbf{E}\tilde{H}_t R_{k-1}\tilde{H}_t
\end{aligned}$$

We have that for diagonal $R_{k-1}$, $\mathbf{x}^\top R_{k-1}\mathbf{x} = \sum_j \lambda_j R_{k-1}[j, j] = Tr(HR_{k-1})$ is deterministic. Therefore,

$$\begin{aligned}
\mathbf{E}\left(\frac{1}{|B|}\sum_{x \in B}\mathbf{x}\mathbf{x}^\top\right)R_{k-1}\left(\frac{1}{|B|}\sum_{x \in B}\mathbf{x}\mathbf{x}^\top\right) &= \left(1 - \frac{1}{|B|}\right)H^2 R_{k-1} + \frac{1}{|B|}\mathbf{E}(\mathbf{x}^\top R_{k-1}\mathbf{x})\mathbf{x}\mathbf{x}^\top \\
&= \left(1 - \frac{1}{|B|}\right)H^2 R_{k-1} + \mathrm{Tr}(HR_{k-1})H \\
&= [\mathbf{E}\tilde{H}_{t-1}^2]R_{k-1}
\end{aligned}$$

Table 3: Second moment statistics for Gauss-Newton-Hessian calculated on ImageNet (IN) on vision and Open-Web-Text-2 (OWT) on language models.

| Model | Size | Data | $\|H\|_{Fr}$ | $\lambda_{\max}(H)$ | $\left(\frac{\|H\|_{Fr}}{\lambda_{\max}(H)}\right)^2$ |
|---|---|---|---|---|---|
| ResNet-18 | 11M | IN | $2.55 \times 10^4$ | $\approx 270$ | $8.92 \times 10^3$ |
| ResNet-50 | 25M | IN | $2.67 \times 10^4$ | $\approx 470$ | $3.22 \times 10^3$ |
| OPT | 1.3B | OWT | $2.94 \times 10^3$ | $\approx 780$ | $1.42 \times 10^1$ |
| Llama-1 | 7B | OWT | $3.73 \times 10^3$ | $\approx 1600$ | $5.43$ |
| Mistral | 7B | OWT | $2.49 \times 10^4$ | $\approx 5600$ | $1.98 \times 10^1$ |

Thus, we conclude

$$R_k = Q^2 R_{k-1} - \eta^2 H^2 R_{k-1} + \eta^2 [\mathbf{E}\tilde{H}_t^2] R_{k-1} = RR_{k-1} = R^k R_0 = R^k.$$

Now, we have that

$$\mathbf{E}\|\mathbf{u}^t\|^2 = [\mathbf{u}^0]^\top R^t \mathbf{u}^0 \geq \overline{\lambda}^t \langle \mathbf{u}^0, v \rangle^2 \to \infty,$$

whenever $\langle \mathbf{u}^0, v \rangle \neq 0$. $\qquad\square$

# D    DETAILS OF HESSIAN STATISTICS CALCULATION

For trace approximation, we use the fact that for a Gaussian vector $\mathbf{g} \in \mathcal{N}(0, \frac{1}{N}I)$, generated independently from $\tilde{H}$, $\mathbf{E}\mathbf{g}^\top \tilde{H}\mathbf{g} = \frac{1}{N}\mathrm{Tr}(H)$. Observe that from (7),

$$\mathbf{g}^\top \tilde{H}\mathbf{g} = \sum_{j=1}^N \left\{ J_\theta h(\mathbf{x};\theta)^\top \mathbf{g} \right\}^\top \{\mathrm{Diag}(\mathrm{sf}(h)) - \mathrm{sf}(h)\mathrm{sf}(h)^\top\} [J_\theta h(\mathbf{x};\theta)^\top \mathbf{g}],$$

thus evaluating such quadratic form can be done by only evaluating the middle Hessian $\mathrm{Diag}(\mathrm{sf}(h)) - \mathrm{sf}(h)\mathrm{sf}(h)^\top$ and finite differences $J_\theta h(\mathbf{x};\theta)^\top \mathbf{g} \approx 50(h(\mathbf{x};\theta + 0.01\mathbf{g}) - h(\mathbf{x};\theta - 0.01\mathbf{g}))$. The results reported in Table 1 are based on 1600 evaluations of $\mathbf{g}_i^\top \tilde{H}_i \mathbf{g}_i$, including the standard error.

At an increased price, we can also evaluate the Frobenius norm of the Hessian. For that, for a given vector $\mathbf{g}$, sampled from Gaussian distribution, we evaluate two independent Hessian-vector products $\tilde{H}\mathbf{g}, \hat{H}\mathbf{g}$, so that $\frac{1}{N}\mathrm{Tr}(H^2) = \mathbf{E}(\tilde{H}\mathbf{g})^\top \hat{H}\mathbf{g}$. Then, by sampling independently a series of independent realizations, we can evaluate the mean $\frac{1}{N}\mathrm{Tr}(H^2)$ and the standard error of our estimation. We report these evaluations in Table 3.

For evaluating the top eigenvalues, one can employ the sketching technique. For example, Swartworth and Woodruff (2023) show that for $\Phi \in \mathbb{R}^{d \times N}$ generated in a way such that $\Phi_{ij} \sim \mathcal{N}(0, \frac{1}{d})$ for $d$ large enough, we have that for finite amount of top eigenvalues,

$$\lambda_l(H) \approx \lambda_l(\Phi H \Phi^\top). \tag{15}$$

In order to evaluate the matrix $\Phi H \Phi^\top$, we iterate over each of $d$ columns $\phi_j$ of $\Phi^\top$ and evaluate the HVP $H\phi$ by sampling empirical $\tilde{H}_t \phi$. For language models, to speed up the caluculation we truncate the context length to 256 and average over batch of size 50. For image classification, we use a batch size of 5000 to evaluate each $H\phi$. We then project each of these columns back with the embedding $\Phi$, so the result is a $d \times d$ matrix, whose top eigenvalue can be calculated with standard linear algebra packages. According to Swartworth and Woodruff (2023), the error term in (15) is bounded by multiple of $\|H\|_{Fr}/\sqrt{d}$. Although we do not offer a precise control, we suggest that taking $d = 5000$ should be sufficient for the models in Table 3. Note that their bound does not account for sampling error and we leave it out of consideration in the scope of this paper, we simply want to produce some adequate bound on the largest eigenvalue.

**Pseudo-random embeddings.** Generating and storing a dense embedding matrix $\Phi \in \mathbb{R}^{d \times N}$ can be prohibitively expensive, since each of the rows $\Phi[i,:]$ is equivalent to one more model in memory. We propose to instead use pseudo-random generators, so that the projector $\Phi$ is generated "on-the-fly" using a single integer number *seed*. The model parameters are usually accessed in a form of lists $\theta^\top = (\theta_1^\top, \ldots, \theta_L^\top)$, where $L$ could be the number of layers. Correspondingly, the gradients and HVPs are also iterated over a list $\mathbf{g} = (\mathbf{g}_1, \ldots, \mathbf{g}_L)$. We generate $\Phi$ in the form $\Phi = (\Phi_1 \quad \ldots \quad \Phi_L)$, so that $\Phi\mathbf{g} = \sum_{j=1}^{L} \Phi_j \mathbf{g}_j$. To calculate $\Phi\mathbf{g}$ we initialize a random generator with a fixed seed, and then generate $\Phi_j$ one after the other so that we never have to store the whole matrix $\Phi$ in the memory.

Nevertheless, the embedding operation itself $\Phi\mathbf{g}$ can be too expensive. Our observation is that starting from $d = 50$ we do not really benefit from parallel matrix computations on GPU and the price scales linearly, i.e. $d = 200$ is 4 times as expensive as $d = 50$, and so on. Furthermore, the price of a single application of the embedding for $d = 50$ can be as high as the gradient computation itself. In order to reduce the computational price of embedding, we suggest to use the following heuristic. Instead of summing up per-layer embeddings $\Phi\mathbf{g} = \Phi_1\mathbf{g}_1 + \cdots + \Phi_L\mathbf{g}_L$, we suggest to concatenate them. This way, we can increase the dimension by a factor of $L$ without overhead computations. Let us denote the resulting concatenating embedding by $\{\mathbf{\Phi}\}$, then we can write

$$\Phi = (\Phi_1 \quad \Phi_2 \quad \ldots \quad \Phi_L) \in \mathbb{R}^{d \times N}$$

$$\{\mathbf{\Phi}\} = \begin{pmatrix} \Phi_1 & 0 & \ldots & \\ 0 & \Phi_1 & & \\ & & \ldots & \\ 0 & 0 & \ldots & \Phi_L \end{pmatrix} \in \mathbb{R}^{Ld \times N}$$

In other words, we have $\{\mathbf{\Phi}\}\theta = \mathtt{vstack}([\Phi_1\theta_1, \ldots, \Phi_L\theta_L])$. That is, each $\Phi_1\theta_1$ has dimension $d$ and

$$\mathbf{E}_\Phi[\{\mathbf{\Phi}\}\theta]^\top[\{\mathbf{\Phi}\}\theta'] = \sum_j \mathbf{E}[\Phi_j\theta_j]^\top[\Phi_j\theta_j'] = \sum_j \theta_j^\top\theta_j' = \theta^\top\theta',$$

so that it preserves the dot products on average as well. However, concatenation can dramatically reduce the variance of one dot product. Indeed, we have that

$$\mathrm{Var}_\Phi([\{\mathbf{\Phi}\}\theta]^\top[\{\mathbf{\Phi}\}\theta']) = \sum_j \mathrm{Var}([\Phi_j\theta_j]^\top[\Phi_j\theta_j']) \lesssim \frac{1}{d}\sum_j \|\theta_j\|^2\|\theta_j'\|^2,$$

and recall the bound from before,

$$\mathrm{Var}_\Phi([\Phi\theta]^\top[\Phi\theta']) \lesssim \frac{1}{d}\|\theta\|^2\|\theta'\|^2 = \frac{1}{d}\left(\sum_j \|\theta_j\|^2\right)\left(\sum_j \|\theta_j'\|^2\right)$$

The former can be much smaller than the latter when square norms of the gradients are spread "evenly" over the layers. That is, assume that $\|\theta_j\|^2$ is approximately in the same bulk $C^{-1}M \leq \|\theta_j\|^2 \leq CM$, $M = \frac{1}{L}\sum_j \|\theta_j\|^2$. Then $\mathrm{Var}_\Phi([\{\mathbf{\Phi}\}\theta]^\top[\{\mathbf{\Phi}\}\theta']) \lesssim \frac{L}{d}M^2$ while $\mathrm{Var}_\Phi([\Phi\theta]^\top[\Phi\theta']) \lesssim \frac{L^2}{d}M^2$, so it is effectively equivalent to increasing the embedding dimension $L$ times compared to the original Gaussian features. Here we ignored the terms $\langle\theta, \theta'\rangle^2$ but in practice they are significantly smaller than $\|\theta\|^2\|\theta'\|^2$.

# E INFLUENCE FUNCTIONS FOR BAG-OF-WORDS MODEL IS A TF-IDF

TF-IDF is a popular measure of word-document relevance used in retrieval systems Ramos et al. (2003). Recall that for a set of documents $d \in D$ and terms $t \in T$, which could be either words or tokens, the term frequency (TF) and document frequency (DF) are defined as follows,

$$TF(t, d) = \frac{1}{|d|}count(t, d),$$

$$DF(t) = \frac{\#\{d \in D : t \in d\}}{|D|},$$

where $count(t, d)$ is the number of occurrences of a term $t$ in document $d$, the latter being a sequence of terms. Let us also consider a variant of inverse document frequency $IDF(t) = \sqrt{1/DF(t)}$. Note

that in standard $TF{\cdot}IDF$ definition, the square root is replaced with logarithm, we only propose to consider the square root for the sake of comparison to influence functions. Then, the document-term relevance $TF \cdot IDF$ is calculated as the product of $TF$ and $IDF$, and the corresponding similarity between $d_1$ and $d_2$ is

$$\text{sim}(d_1, d_2) = \sum_t TF \cdot IDF(d_1, t)TF \cdot IDF(d_2, t) = \sum_t \frac{1}{DF(t)} TF(d_1, t)TF(d_2, t)$$

The bag-of-words model reads as follows,

$$\log p(d) = \sum_t count(t, d) \log p_t,$$

where $p_t = \exp(x_t)/(\exp(x_1) + \cdots + \exp(x_T))$, the $x_t$ are parameters. For the sake of simplicity we assume that each document has the same length $|d|$. We have,

$$\nabla_x \log p(d) = \sum_t count(t, d)e_t - |d|\nabla_x \log \left( \sum_t \exp(x_t) \right) = \sum_t count(t, d)e_t - |d|\text{sf}(x)$$

$$= |d| \left\{ \sum_t TF(t, d)e_t - \text{sf}(x) \right\} \tag{16}$$

Notice that by definition, $\mathbf{1}^\top \nabla_x \log p(d) = 0$. The Hessian looks as follows

$$\nabla_x^2 \mathbf{E} \log p(d) = -|d|\nabla_x^\top \text{sf}(x) = -|d| \left( \text{Diag}(\text{sf}(x)) - \text{sf}(x)\text{sf}(x)^\top \right),$$

where we calculate that

$$\nabla_{x_i} \frac{\exp(x_j)}{\sum_k \exp(x_k)} = \frac{\delta_{ij} \exp(x_i)(\sum_k \exp(x_k)) - \exp(x_i) \exp(x_j)}{(\sum_k \exp(x_k))^2} = \delta_{ij}\text{sf}(x)_i - \text{sf}(x)_i\text{sf}(x)_j$$

Let us calculate the inverse Hessian. Given a damping parameter $\lambda$, let $\mathbf{q} = (\mathbf{p} + \lambda)^{1/2}$ elementwise. Note that $\|\mathbf{q}\|^2 = 1 + \lambda N$. Set also $\mathbf{r} = \mathbf{p}/(\mathbf{p} + \lambda)^{1/2}$ elementwise, and notice that $\|r\|^2 = \sum_j p_j^2/(p_j + \lambda) = 1 - \lambda \sum_j p_j/(p_j + \lambda) < 1$. Then,

$$\text{Diag}(\mathbf{p}) + \lambda I - \mathbf{p}\mathbf{p}^\top = \text{Diag}(\mathbf{q})(I - \mathbf{r}\mathbf{r}^\top)\text{Diag}(\mathbf{q}),$$

so the inverse equals to

$$(H + \lambda)^{-1} = \text{Diag}(\mathbf{q})^{-1} \left( I + \frac{1}{1 - \|\mathbf{r}\|^2} \mathbf{r}\mathbf{r}^\top \right) \text{Diag}(\mathbf{q})^{-1}$$

$$= \text{Diag}(\mathbf{p} + \lambda)^{-1} - \left( \lambda \sum_j p_j/(p_j + \lambda) \right)^{-1} \mathbf{d}\mathbf{d}^\top,$$

where we denote $\mathbf{d} = \mathbf{p}/(\mathbf{p} + \lambda) = \mathbf{1} - \lambda/(\mathbf{p} + \lambda)$ elementwise. Note that the gradients $\mathbf{g}_d = \nabla \log p(d)$ are in the subset $\mathbf{g}_d^\top \mathbf{1}$. Since $\mathbf{d} = \mathbf{1} + O(\lambda)$ we have that

$$\mathbf{g}_{d_1}^\top \left( \lambda \sum_j p_j/(p_j + \lambda) \right)^{-1} \mathbf{d}\mathbf{d}^\top \mathbf{g}_{d_2} = O(\lambda),$$

therefore, in the limit $\lambda \to 0$, we have that the influence between documents $d_1, d_2$ reads as

$$\mathcal{I}(d_1, d_2) = \mathbf{g}_{d_1}^\top \text{Diag}(\mathbf{p})^{-1}\mathbf{g}_{d_2} = \sum_t TF(t, d_1)TF(t, d_2)p_t^{-1}.$$

Let us also calculate the IDF for this models. Let us assume that terms are rare enough ( $p_t|d| \ll 1$). Then we have that,

$$DF(d) = 1 - \Pr(t \notin d) = 1 - (1 - p_t)^{|d|} \approx |d|p_t.$$

and therefore up to a scaling factor, the above expression is approximately equal to the $TF \cdot IDF$ relevance in (16).

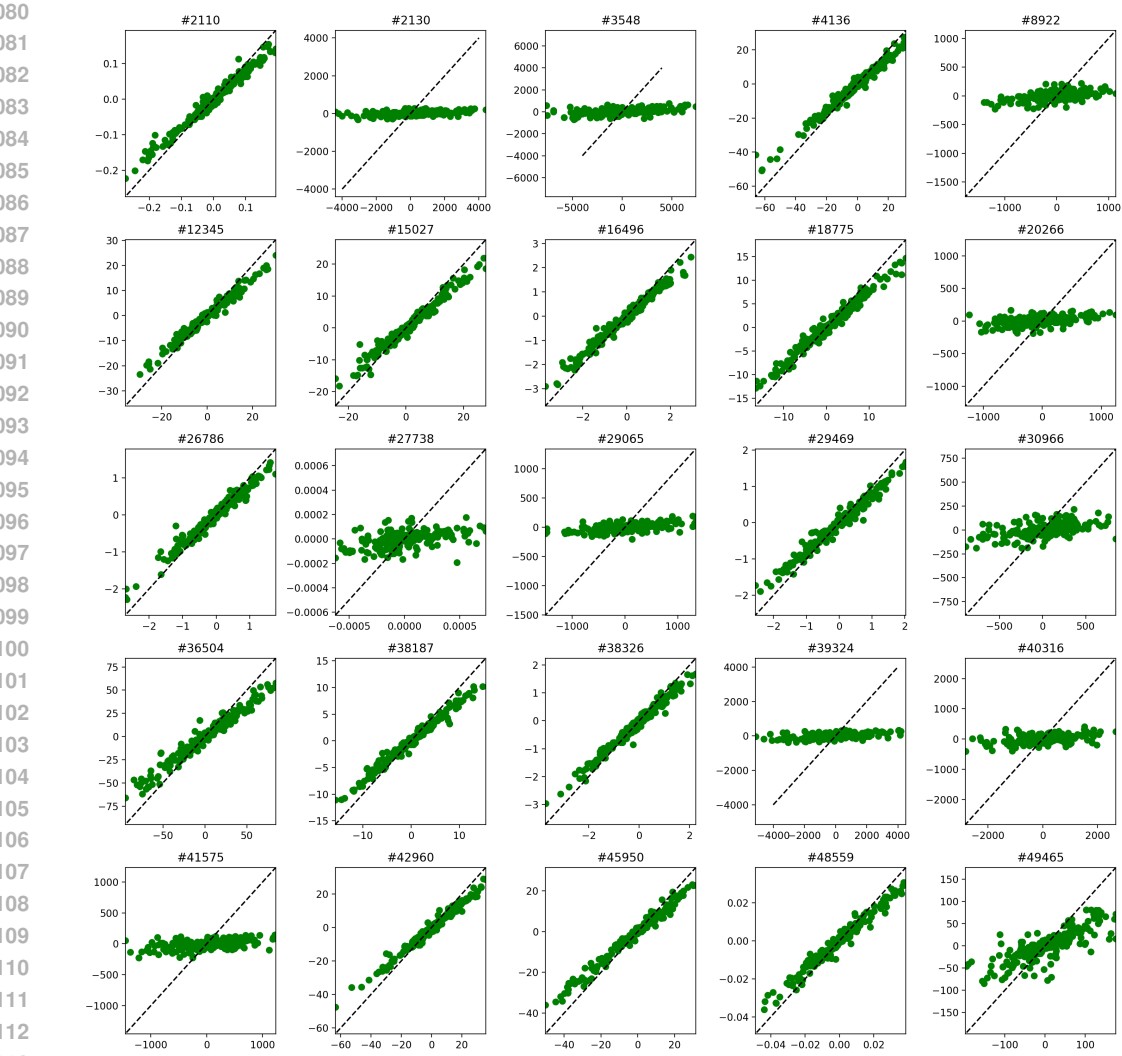

Figure 6: Comparison of the PBRF and LiSSA influence on ResNet-18 for 25 random train images. Each graph shows influence of one train image w.r.t. to 500 other test images. Reference number is show above the image, refer to Figure 8. The results are for ResNet-18, the $x$-axis is the LiSSA, and the $y$-axis is the PBRF.

# F    MORE EXAMPLES FOR COMPARISON OF THE LISSA AND PBRF

Here we present a complete list of 25 train images for comparison of the PBRF and LiSSA. We calculate the LiSSA according to the hyperparameter recommendations in Table 1. Figure 6 shows scatter plots of the LiSSA and PBRF for ResNet-18, and Figure 7 shows scatter plots for ResNet-50. Figure 8 shows reference images from the ImageNet dataset. Note that for one train image in Figure 7 the LiSSA got float overflow, we attribute it to the high value of the gradient norm.

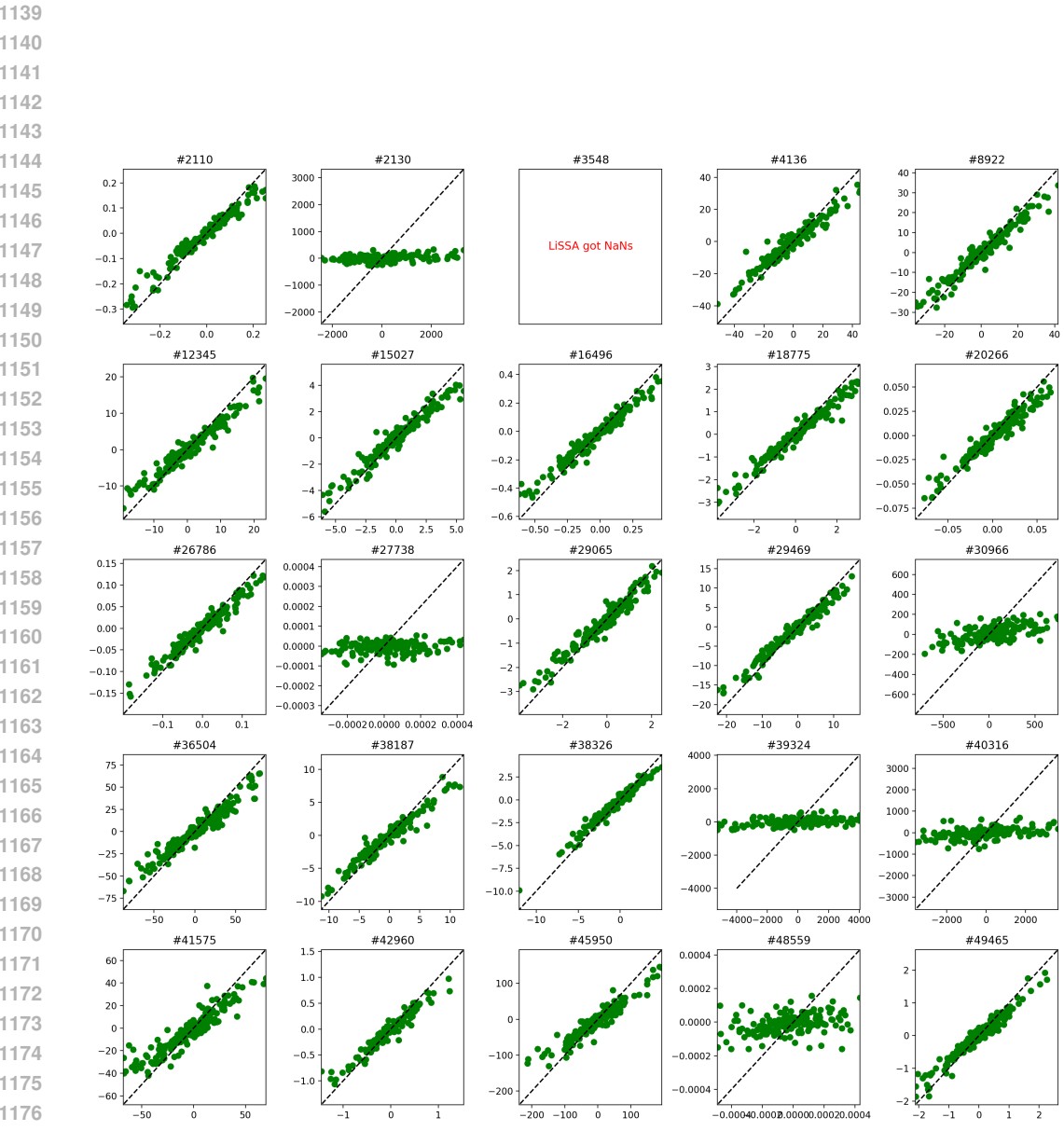

Figure 7: Same as Figure 6, but for ResNet-50.

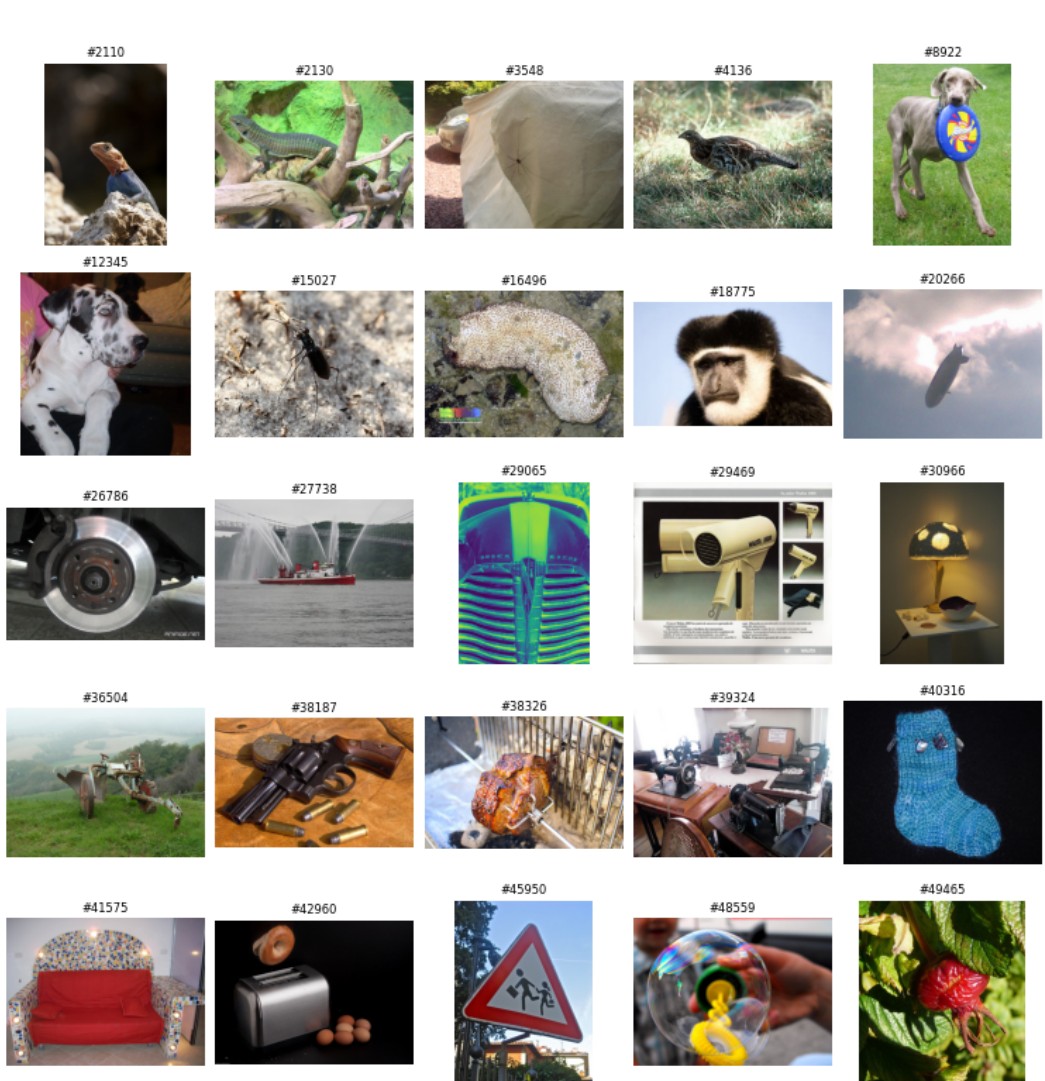

Figure 8: Reference images from ImageNet. Numbers above each image corresponde to those Figures 6, 7.

