# OpenReview forum: "Revisiting inverse Hessian vector products for calculating influence functions"
_ICLR.cc/2025/Conference — Submitted to ICLR 2025_

### Official Review · Reviewer_1Abi · 2024-10-27

**Soundness:** 3
**Presentation:** 3
**Contribution:** 2
**Rating:** 6
**Confidence:** 4

**Summary:**

The paper considers inverse Hessian-vector products for the purpose of evaluating the influence of training data $(x_m, y_m)$, a data-label pair, on fitted parameters $\theta^*.

For application to empirical problems, in particular language models, the Hessian is replaced by the Gauss-Newton matrix, and a regularized inverse-Hessian product is considered $(\lambda + H)^{-1}$. Extensive details are provided on the numerical considerations taken in evaluating Hessian-vector products.

An algorithm, LISSA, for estimating the Hessian-vector-product on a training point is considered.  It is effectively a version of minibatch SGD on a quadratic problem (8), where the gradient estimator is specific to this problem instance.  A convergence analysis in expectation and norm-convergence (up to a neighborhood) is provided.  These are taken under a relatively strong norm-type assumption on the minibatch gradients (C.1).

This norm-type assumption is discussed in theory in the appendix in two settings and also empirically.

A table of hyperparameters are provided for a range of reasonable setups (Table 1), which (appear to be) conforming to the assumption C.1.  These require estimating the trace and operator norm of the hessian, and some discussion of the sketching procedure taken to do this is provided.

The convergence of LISSA is tested with these hyperparameters, to show good performance, and the influence is measured against a second method (PBRF).

Finally some discussion of the value of inverse-Hessian influence (vs gradient-based influence measures) are provided in Section 5.

**Strengths:**

The paper does well to describe the end-to-end process of using inverse-Hessian influence metric on LLMs and Resnets, for which there are substantial empirical and theoretical questions to address.  It would appear that the cases that the authors considered are evaluated in reasonable level of detail, and that the empirical aspects of the paper should be reproducible.

While the main contributions are about methodology of using inverse-Hessian influences; section 5 provides some additional scientific value on the potential value of using inverse-Hessian influences.  While it is more speculative in nature, it provides some problem setups which might be the setting for future work.

The paper is also well-integrated into existing theory on influence functions, which brings additional value to the paper.

The main theorem seems to provide a suggested starting point for how to pick the parameters in LISSA (with explicit parameters).

**Weaknesses:**

I find the following overall weaknesses:

1) The main theoretical contribution of the paper is a convergence analysis for the mini-batch SGD derived.  This is done under a very strong hypothesis, which is effectively a contractivity assumption in norm (C.1) or (9) in Theorem 1.  This makes the main theorem relatively easy to prove, but it also begs the question of showing when this satisfied.   Separately, mini-batch SGD on quadratics are studied to death, and it will be quite difficult to argue that this is truly a new analysis

2) Regarding the contractivity assumption: it is shown that in some sense the assumptions of Theorem 1 necessary, but this sense is effectively adversarial in the test-direction u (B.4Lemma 1).     On the other hand, the positive cases (B3 Motivations for C.1) do not show that C1 is satisfied.  It would substantially improve the paper if the theory covered these two cases.  (If it does, this should be explained).

3) The empirical results of the contractivity (Figure 4) do not consider C.1 but rather the trace.  Trace type conditions are much easier to satisfy, and unfortunately give little insight whether C.1 in fact holds.  A much more desirable theorem (to my taste -- and also more consistent with the experiments) would assume trace-type conditions on H and H^2, and some randomness assumptions on u that lead to the claimed convergence.  There is a lingering sense that C.1 and (9) of Theorem 1 is unverifiable.

4) The empirical conclusions in Section 4 and 5 are a little bit murky, and it's hard to align what appears in the simulations with what appear in the text.

**Questions:**

Regarding the weaknesses:
Weakness 1): There's a lot of batch-SGD analyses which at first glance may already contain the theorem.  For example:
A) Jain, Kakade, Kidambi, Netrapalli, Sidford. "Parallelizing.." 2018 JMLR
B) Varre, Pillaud-Vivien, Flammarion. "Last iterate convergnece.." 2021 NeurIPS
C) Ma, Bassily, Belkin.  "The power of Interpolation..." 2018 ICML
Can you argue why your analysis is novel, comparing to existing literature like the above?

Weakness 2): Can you show why the theorems apply to the setups in B3? Currently, this is `almost' done but not completely.  In particular the sentence on (l243): ``we find this condition to hold for classification with independent sampling" appears unfulfilled."

Weakness 3): Can you show empirically that C1 occurs, in the norm sense?  (tracial vs norm-type behavior is rarely equivalent in random matrix contexts).

Weakness 4a): Figure 1 is appears to be discussed in the text (l368-370) and 3 cases are explained. Considering Figure 1, why are these the 3 cases, (especially, which illustrates case '3'? Where do you argue that 'PBO _finetunning_ stirs away the model too far for the quadratic approximation to hold'. (and what does that mean?)

Weakness 4b): Figure 3.  Why does the difference picture show what you claim it does?  (In particular why do you average over the 10x10 square, and why are you not measuring the diagonals?). When you write (l464-466). ``As a result the influence similarity between an orignal sentence and a rewritten one appears to be consistently higher than between unrelated sentence."  Can you explain this?

Other questions:
1) Can you explain precisely how you picked the 'batch' in Table 1, and moreover, what is the general procedure you suggest for selecting LiSSA parameters?  (It seems you _almost_ do this, but being extra-clear would help, especially as regard the batch.)

Minor:
1) (l80-83). ``Influence functions are calculated under the assumption that the optimized parameter $\theta$ of the model delivers minimum to the training loss."  This sentence is mangled.
2) l409-410.  ``the choice of _dumping_ parameter''
3) l424: ``For example, a plausible interpretation of the directions $v_j$ would be that the top directions correspond to general language coherence and... small eigenvalues could correspond to more specific, informative content."  While the statement about large eigenvalues makes sense, the complementary statement about small eigenvalues does not -- first most small eigenvalues are not interpretable, and are the result of either diffuse (non-one-dimensional) embeddings or undesirable feature distortions.  Perhaps it would be better to just say you can screen out the large eigenvalues which are more about general language coherence.

---

> ### Author Response · Authors · 2024-11-21
> **Response to reviewer 1Abi**
>
> We thank the reviewer for valuable feedback. We hope that the revised version can address the reviewer's concerns. Below we address each question separately.
>
> 1) *"There's a lot of batch-SGD analyses which at first glance may already contain the theorem. For example: A) Jain, Kakade, Kidambi, Netrapalli, Sidford. "Parallelizing.." 2018 JMLR B) Varre, Pillaud-Vivien, Flammarion. "Last iterate convergnece.." 2021 NeurIPS C) Ma, Bassily, Belkin. "The power of Interpolation..." 2018 ICML Can you argue why your analysis is novel, comparing to existing literature like the above?"*
>
> Indeed, it is an oversight to our side, the analysis of linear regression in Ma, Bassily, Belkin is pretty much the same as ours. In the revised version, we moved Theorem 1 to the appendix as part of the proof of the Corollary (which is now called Theorem). We do so to focus more on the condition C.1 which is missing from the previous literature. In particular, Ma, Bassily, Belkin do not provide an expression for critical batch size that depends solely on Tr(H) and lambda_max(H). In addition, we provide a counter example that shows if the batch size condition is broken, the algorithm does not observe. In our counter example, H can be arbitrary and the condition C.1 is met. Ma, Bassily, Belkin provide a similar counter example but only for the case H = cI. The results in *Jain, Kakade, Kidambi, Netrapalli, Sidford. Parallelizing..* and also *Dieuleveut et al. Harder, Better, Faster, Stronger Convergence Rates for Least-Squares Regression* are slightly different, they allow a reduced learning rate compared to optimal step in the deterministic case $\eta = 1 / \lambda_{\max} $
>
>
> 2) *"Can you show why the theorems apply to the setups in B3? Currently, this is `almost' done but not completely."*
>
> In the revised version, we make a precise statement in the main part of the text.
>
> 3) *"Can you show empirically that C1 occurs, in the norm sense? (tracial vs norm-type behavior is rarely equivalent in random matrix contexts)."*
>
> We find it is rather difficult to evaluate the norm of $\tilde{H}_t^{2}$. We compute a relatively low-dimensional random projection ($d = 96$) of the difference of RHS and LHS for a small GPT2 model, results shown in Figure 5 in the revised version
>
> 4) *"Figure 1 is appears to be discussed in the text (l368-370) and 3 cases are explained. Considering Figure 1, why are these the 3 cases, (especially, which illustrates case '3'? Where do you argue that 'PBO finetunning stirs away the model too far for the quadratic approximation to hold'. (and what does that mean?)"*
>
> Simply because the values of the influence are much larger than in rest of the pictures. The locality in "..quadratic approximation" depends on epsilon in formula (5), but we also cannot take it too small, since we can encounter problems with floating point.
>
> 5) *"Figure 3. Why does the difference picture show what you claim it does? (In particular why do you average over the 10x10 square, and why are you not measuring the diagonals?). When you write (l464-466). ``As a result the influence similarity between an orignal sentence and a rewritten one appears to be consistently higher than between unrelated sentence." Can you explain this?"*
>
> Here, the diagonal (at positions [0, 10], [1, 11], ... etc) correspond to the similarity of a sentence and it's paraphrased version, while the off-diagonal elements correspond to similarity of completely unrelated sentences.
>
> 6) *"Can you explain precisely how you picked the 'batch' in Table 1, and moreover, what is the general procedure you suggest for selecting LiSSA parameters? (It seems you almost do this, but being extra-clear would help, especially as regard the batch.)"*
>
> The batch is chosen based on Theorem 1's condition $|B| >= C Tr(H) / \lambda_{\max}(H)$; we take $C = 2$ conventionally. This requires evaluation trace and largest eigenvalue. Trace is rather trivial, through random gaussian quadratic form, we can even calculate the std which is shown in Table 1. Largest eigenvalue is more complicated. We use sketches (Swartworth and Woodruff), however the quality of the approximation depends on the dimension, and the $Tr(H^2)$. We select $d = 5000$ based on some evaluations in the appendix (lines 963-971) However, we do not have a rigorous way to evaluate error of estimation for lambda_max.
>
> We thank the reviewer for minor corrections, we make the corresponding changes in the revised version.

---

> > ### Comment · Reviewer_1Abi · 2024-11-23
> >
> > Thank you for the clarifications and the revisions in the manuscript.  I don't have any additional questions.
> >
> > I believe there are plenty of contributions here, especially the examples and the empirical work, which justify the acceptance of the paper.  So I have raised my score.
> >
> > I'll add -- perhaps directed at my fellow reviewers -- that Theorem 1, while simple, is just 1 part of the paper.  And the authors have arrived at useful batch and step-size choices.  That is a lot more than can be said for 90% of the ML optimization theory -- even if the assumption C1 is strong.
> >
> > (Heads up -- I see you still use the words "dumping parameters"... For what it's worth, this reviewer enjoys it.)

---

> > > ### Author Response · Authors · 2024-11-25
> > >
> > > Thanks a lot for your feedback, we really appreciate it.
> > >
> > > All the "dumping parameters" have been corrected in the last revised version, thanks.

---

### Official Review · Reviewer_fS9T · 2024-11-01

**Soundness:** 3
**Presentation:** 3
**Contribution:** 3
**Rating:** 6
**Confidence:** 4

**Summary:**

This paper studies influence functions, which are a popular tool for attributing models' outputs to training data. This reduces to calculation of inverse Hessian-vector products (iHVP) using the classical "Linear Time Stochastic Second-order Algorithm" (LiSSA) method. The authors show that the two specific spectral properties of the Hessian, namely its trace and largest eigenvalue, determines the hyper parameters like the scaling factor, the batch size and the number of steps. So they evaluate these two quantities using random sketching. The conclusion is that batch size has to be sufficiently large for the LiSSA to converge.

**Strengths:**

- The authors show that the two specific spectral properties of the Hessian, namely its trace and largest eigenvalue, determines the hyper parameters like the scaling factor, the batch size and the number of steps. So they evaluate these two quantities using random sketching. The conclusion is that batch size has to be sufficiently large for the LiSSA to converge.

- code is open-source shared

**Weaknesses:**

Overall I think the paper looks good.

**Questions:**

Since this topic is about hyerparameter tuning, can we connect the proposed theory/results with iterative gradient-based optimization methods? More specifically, I know that for optimization algorithms like GD, its behavior will be influenced by the Hessian near equilibrium (i.e., the condition number of Hessian). So I am curious if this phenomenon is related to what is proposed in this paper.

---

> ### Author Response · Authors · 2024-11-21
> **Response to reviewer fS9T**
>
> We thank the reviewer for feedback. We address the question in the review below:
>
> *"Since this topic is about hyperparameter tuning, can we connect the proposed theory/results with iterative gradient-based optimization methods? More specifically, I know that for optimization algorithms like GD, its behavior will be influenced by the Hessian near equilibrium (i.e., the condition number of Hessian). So I am curious if this phenomenon is related to what is proposed in this paper."*
>
> This is an interesting point, thank you for bringing this to our attention. Perhaps you are referring to "edge of stability" https://arxiv.org/pdf/2103.00065 ? It seems it is not that straightforward to make the connection to our work, the optimisers that are used to train neural nets are more complicated than the plain SGD.

---

> > ### Comment · Reviewer_fS9T · 2024-11-21
> > **response**
> >
> > Sure. I was just talking about the very basic fact that say methods with momentum like Polyak or Nesterov can behave better as compared to vanilla GD when the Hessian's condition number is large.

---

> ### Comment · Reviewer_fS9T · 2024-12-02
> **keep the score**
>
> rt

---

> > ### Author Response · Authors · 2024-12-02
> >
> > We appoligize for the delayed response and thank you for the reminder.
> >
> > There appears to be a connection to a popular problem of critical batch size. For instance, Ma, Bassilly, Belkin, study linear regression problem very similar to ours. Their goal is slightly different however. They consider how choice of batch affects number of steps of SGD required to achieve certain loss. In their case, there is a critical batch size, such that for $m < m*$ one can adjust the learning rate and the number of steps proportionally to achieve the same accuracy, however the total amount of compute required does not change within that range, meaning we can benefit from parallel computations. For $m > m^{*}$ this does not quite happen as we can no longer increase the learning rate and decrease the number of steps.
> >
> > In our formulation, we do the opposite, we take the learning rate  that corresponds to the least number of steps $ \eta = 1 / (\lambda_{\max}(H) + \lambda)$  according to the previous literature (Koh and Liang). Then, the inequality reverses, we have to choose a batch size that is sufficiently large. In principle, we could reduce the batch size and the learning rate $\eta$ proportionally, and this would correspond to the "linearly scaling" regime of Ma, Bassily and Belkin.
> >
> > We note that both us and Ma, Bassily, Belkin study a plain SGD algorithm (LiSSA can be interpreted as one). Analysis of momentum methods seems to only appear in empirical studies. Kaplan et al study scaling laws for Adam (see formula 5.3) and C. Shallue et al  find that momentum SGD can benefit from even larger batches. Also more recent paper H Zhang et al.
> >
> >
> > Ma, Bassilly, Belkin. (2018) The Power of Interpolation  https://arxiv.org/pdf/1712.06559
> >
> > J. Kaplan et al. (2020) Scaling Laws for Neural Language Models
> >
> > C. J. Shallue et al. (2019) Measuring the Effects of Data Parallelism on Neural Network Training
> >
> >
> > Hanlin Zhang et al (2024) How Does Critical Batch Size Scale in Pre-training?

---

### Official Review · Reviewer_qCYo · 2024-11-03

**Soundness:** 2
**Presentation:** 2
**Contribution:** 2
**Rating:** 5
**Confidence:** 4

**Summary:**

The submission presents an analysis of the convergence of SGD for solving the inverse Hessian-vector-product, useful in the context of computing influence functions. The submission presents convergence results that depend on statistics of the Hessian and the batch size, and show that using those to set hyperparameters to achieve a desired accuracy leads to good performance, automating the choice of hyperparameters.

**Strengths:**

That the convergence theory for SGD can automate hyperparameter selection is a nice observation that is likely to be helpful to the community.

**Weaknesses:**

The main weakness is the first contribution of the submission, the convergence analysis in section 3. The presentation of the results ignores previous work on stochastic optimization and the description of the results lacks clarity. The main novelty would be the boxed condition C.1. However, it is unclear why this bound holds, as there is no theoretical explanation nor empirical validation provided in the main text. See detailed comments in Questions below.

**Questions:**

**Issues with the theoretical results**
- The submission does not address the large optimization literature on stochastic gradient descent, which has derived many similar convergence guarantees. The rates are not typically written in terms of a linear + constant term, as this is not convergent, and the literature is typically concerned with decreasing step-sizes (see e.g. [Bach and Moulines, 2013](https://arxiv.org/pdf/1306.2119)) or other schemes that can guarantee convergence. But the linear + constant result still appears regularly, for example see Theorem 2.1 in the work of [Needell et al., 2015](https://arxiv.org/pdf/1310.5715), Theorem 1 in the work of [Dieuleveut et al., 2016](https://arxiv.org/pdf/1602.05419), of Theorem 5.8 in the handbook of [Garrigos and Gower](https://arxiv.org/pdf/2301.11235). Given those results, the additional insights provided by the current submission appears small, but I might have missed a point that is specific to the inverse Hessian-product setting. If so, a discussion of existing literature in optimization and highlighting what makes the iHVP problem special would make the submission stronger.

- The text uses the word "converges" loosely, e.g. in corollary 1, "the algorithm converges in $T = \Omega(1/(\eta\lambda))$ steps". Taken literally, this is wrong, as the algorithm is not guaranteed to return $u^t = u^*$ after $T$ steps. I assume this is meant to say that the approximation error is bounded by some quantity after $T$ steps, but it isn't clear from the current writing. Changing the statement to a typical statement of "to achieve an error of $\frac{1}{2}\| u^t - u^*\|^2 \leq ...$, the algorithm should be run with parameters $\eta, B = ...$ for $T \geq ...$ iterations" would make it clearer. If the number of iterations is an upper bound, tt should also be $O$ rather than $\Omega$.

- The condition C.1 seems convenient but it isn't clear from the main text why it is introduced, or why it should hold. Extracting the key argument from the appendix to the main text to provide an explanation for the condition and why it is expected to hold would help readers understand the contribution. That the covariance scales with the trace appears to imply/assume that the covariance and the Hessian are the same, as in the toy quadratic model of [Zhang et al.](https://proceedings.neurips.cc/paper/2019/file/e0eacd983971634327ae1819ea8b6214-Paper.pdf) (see §3.5), and a citation to other people using similar assumptions might help make the case.

- The text "confirms the inverse scaling with batch size". Since the LHS of (C.1) is a variance term, the scaling as $\propto 1/B$ does not seem to need confirmation. The assumption that it is scaling with the trace however does seem to warrant empirical evidence. The scaling with the trace of the Hessian is the less intuitive part of the condition, and adding evidence for the scaling would strengthen the claim.

**Writing**
The writing would benefit from an additional pass to improve the clarity of the message. Some of the statements appear wrong or misleading and the text contain typos or broken sentences. I only list a few examples below.

- Re: "Koh and Liang (2017) proposed a stochastic iterative approach called [...] (LiSSA, (Agarawal et al. 2017))". A more accurate description would be that Koh and Liang propose to _use_ the LiSSA algorithm of Agarawal et al., rather than propose the algorithm.
- Description such as "Basu et al. (2020) criticize the LiSSA [algorithm], suggesting that it lacks convergence for deep networks" and "we carefully analyze the convergence of the Lissa [algorithm]" should be made more specific. It is not clear whether this is refering to theoretical properties (will the algorithm actually converge to the solution if run sufficiently long) or whether the term "convergence" is used to refer to empirical performance, closer to "suggesting that the method performs poorly on deep networks" and "we analyze the [empirical performance] of the LiSSA algorithm as a function of its hyperparameters".
- The sentence "SGD is known to work at least as well as the full gradient descent, even in terms of the number of updates (Harvey et al., 2019)" is misleading. Harvey et al. treat the non-smooth convex case, which is significantly different from the quadratic case considered here. The sentence should be removed.
- The equality in Eq. (3) should be replaced by an $\approx$ as it is the result from a truncated Taylor expansion
- L27-28, "interpret the internal computations in an understandable to a human way"
- "Generally speaking this is different from the empirical FIM [...], however, for low noise distributions the two might be used interchangeably (Martens, 2020)". I do not recall Martens making this argument, and it is not clear what is meant by "low-noise distribution".

---

> ### Author Response · Authors · 2024-11-21
> **Response to reviewer qCYo**
>
> We are grateful to the reviewer for elaborate comments and suggestions. Below we address each comment separately.
>
> 1) *"The submission does not address the large optimization literature on stochastic gradient descent, which has derived many similar convergence guarantees. The rates are not typically written in terms of a linear + constant term, as this is not convergent, and the literature is typically concerned with decreasing step-sizes (see e.g. Bach and Moulines, 2013) or other schemes that can guarantee convergence. But the linear + constant result still appears regularly, for example see Theorem 2.1 in the work of Needell et al., 2015, Theorem 1 in the work of Dieuleveut et al., 2016, of Theorem 5.8 in the handbook of Garrigos and Gower. Given those results, the additional insights provided by the current submission appears small, but I might have missed a point that is specific to the inverse Hessian-product setting. If so, a discussion of existing literature in optimization and highlighting what makes the iHVP problem special would make the submission stronger."*
>
> Thank you for pointing this literature out. We find that general optimization results often are not sufficient for accurate analysis. In particular, among the papers that you have mentioned, only Theorem 1 considers a form of average in-batch smoothness, however, they require learning rate to be as small as $ 1/ Tr(H)$ which is much smaller. Our goal is to match the standard choice $\eta = 1/\lambda_{\max}(H)$ with just the right batch size. Tang et el (2020) that we were already citing requires a weaker condition, a bound on $\mathbf{E} || \tilde{H}_t ||^{2}$, which also would require reducing the learning rate. The closest to our work is Ma, Bassily, and Belkin The Power of Interpolation (2018) https://arxiv.org/pdf/1712.06559, in particular their proof of the linear regression case is indeed very close to ours. However, they do not have an equivalent of condition C.1, instead they require a uniform upper bound on the norm of features (equivalent to our gradients). For this reason, we decided to put more emphasis on the condition in the revised version. We also note that we also include a lower bound that adds more value compared to Ma, Bassily, and Belkin.
>
> 2) *"The text uses the word "converges" loosely,"*
>
> Corrected, where possible. In some cases we have to use it loosely, such as in the experiment in Figure 2.
>
> 3) *"The condition C.1 seems convenient but it isn't clear from the main text why it is introduced..."*
>
> We've put more emphasis on this condition in the main text, stating it as a lemma under some relatively weak distributional condition in classification case + incoherence condition in the language modelling case. We do not quite understand how we can compare to Zhang et al . They compare Hessian with covariance of the gradient. While we compare Hessian with average squared-batched-Hessian. I believe the confusion comes from the fact that the model in Zhang et al does not include the linear part, which would typically be associated with gradients.
>
> 4) *"The text "confirms the inverse scaling with batch size".  ... adding evidence for the scaling would strengthen the claim."*
>
> We note that the scaling with trace is confirmed by considering different models with different traces in Figure 4. To strengthen this argument, we added comparison of projected LHS and RHS as matrices of dimension 96 x 96. This is rather expensive experiment and we only consider a small GPT2 model, which incidentally corresponds to the dependent sampling case.
>
> 5) *"Re: "Koh and Liang (2017) proposed ..."*
>
> Corrected, thank you
>
> 6) *"Description such as "Basu et al. (2020) criticize the LiSSA [algorithm], suggesting that it lacks convergence for deep networks" and "we carefully analyze the convergence of the Lissa [algorithm]" should be made more specific..."*
>
> Thank you for pointing this out. Indeed, it sounds vague and we made it more specific in the revised version.
>
> 7) *"The sentence "SGD is known to work at least as well ... should be removed."*
>
> Indeed, this paper is not relevant. We removed this sentence from the revised version.
>
> 8) *"The equality in Eq. (3) should be replaced by an  as it is the result from a truncated Taylor expansion"*
>
> This is equality for the derivative, which is taken at epsilon = 0
>
> 9) *"L27-28, "interpret the internal computations in an understandable to a human way"*
>
> Added "of neural networks"
>
> 10) *"I do not recall Martens making this argument, and it is not clear what is meant by "low-noise distribution".*
>
> Thank you for this comment. We changed the reference to *Kunstner et al. Limitations of the empirical fisher approximation for natural
> gradient descent*; we also changed "low-noise" to "realizable", the term they use in Section 4.2. By low-noise we mean a model that converged to $p(y|x) \approx 1$ on the training data, which is better described as realizable.

---

> > ### Comment · Reviewer_qCYo · 2024-11-26
> >
> > I thank the authors for their response.
> >
> > I agree with reviewer 1Abi that the submission has value, but I am still worried about overly strong claims that run counter to the optimization literature. As section 3 is the core motivation for the proposed hyperparameters, it could mislead readers into what the method is achieving. Given the extension to the discussion period, I hope the following concerns can be adressed. An edit to the submission is not necessary, but I would like to know whether the authors agree/disagree with those concerns.
> >
> > First, to check my understanding, would the authors agree that the following is a fair description of the contribution in §3?
> >
> > > This section derives a bound on the expected squared error that decomposes into a bias and variance terms (called convergence and sampling errors in the submission) that, for the step-size $\eta=1/L$ and a specific choice of batch size that allows one to recover the classical deterministic strongly-convex convergence rate of $1-\lambda \eta$ with $\eta = 1/\lambda_{\max}(H)$ _on the bias term_.
> >
> > If this is correct, my main concern is that this property, which seems to be refered to as "convergence", does not imply convergence in the sense typically found in optimization. My understanding is that the analysis takes the step-size $1/L$ ($L=\lambda_{\max}+\lambda$) for granted and selects the batch size to make the optimization "not diverge". But those hyperparameters should be phrased as heuristics, as the step-size need not be optimal and the batch-size selected does not ensure convergence.
> >
> > That the submission proposes a heuristic to select hyperparameters is a good thing; my issue is that it appears to be framed as ensuring convergence when it does not.
> >
> > ---
> >
> > > they require learning rate to be as small as [] which is much smaller. Our goal is to match the standard choice [] with just the right batch size.
> >
> > On the step-size. While "larger step-sizes are better" is intuitive, this is not clear in stochastic optimization as we need to reduce the noise to converge.
> > Either by reducing the step-size or averaging, as in the papers previously mentioned, [increasing the batch size](https://arxiv.org/abs/1104.2373), or through [variance reduction methods](https://arxiv.org/abs/1309.2388).
> >
> > By convergence, the optimization literature typically means $\mathbb{E}[f(x_t) - f(x_*)] \to 0$ as $t \to \infty$ (or some other measure of error). This does not happen for the proposed hyperparameters as a constant step-size and constant batch-size will hit a noise ball. Alternatively, one could talk of convergence to some $\epsilon$ in a given budget, but Theorem 1 does not provide this as the variance term depends on the stochastic gradients, which implicitly depends on $u_t$, and is not controlled. As a result, it is indeed difficult to directly compare the results of this submission with existing results in the literature, as they typically try to ensure one of the two conditions above.
> >
> > ---
> >
> > Below are what I think needs to be changed for the paper to accurately represent the contribution
> >
> > - The submission should explicitly acknowledge that
> >   - The proposed method is a heuristic, and that the method will not converge.
> >     It is perfectly fine to argue for the proposed heuristic by validating it experimentally and showing that the achieved precision is sufficient for the purposes of computing influence functions.
> >   - The choice of $1/L$ as the step-size is arbitrary, and that other step-sizes might work better if the goal is to ensure convergence.
> >     It is fine to motivate the proposed heuristic by "using the step-size that works for the deterministic case", my issue is that the current phrasing implicitly assumes that $1/L$ is optimal for the current setting.
> >
> >
> > - The following statements regarding convergence should be changed.
> >
> >   > we find that the batch size has to be sufficiently large for the LiSSA to converge (abstract)
> >   > we find that the batch size has to be sufficiently large for the algorithm to converge (p1)
> >   > and we can say that it measures how quickly we converged to the solution (p5)
> >
> >   I understand that "the algorithm does not diverge" is less appealing than "the algorithm converge".
> >   I would suggest saying that the optimization is "stable" and explicitly defining a condition like "Stable step-size/batch-size pair" in the text.
> >
> > - The discussion of the optimization literature should make clear that the objective is typically different (ensuring convergence), and therefore leads to different decisions.
> > work of Belkin et al as "not mak[ing] clear connection to the trace of the Hessian in determining critical batch size" is technically correct as the "critical batch size" as defined by the submission is not present in their work but that statement appears disingenuous as that notion of critically is not typically relevant for optimization that try to achieve convergence as the step-size has to be smaller anyway to counteract the noise.

---

> > > ### Comment · Reviewer_qCYo · 2024-11-26
> > >
> > > Minor points (does not require a response)
> > >
> > > > > "The equality in Eq. (3) should be replaced by an as it is the result from a truncated Taylor expansion"
> > > >
> > > > This is equality for the derivative, which is taken at epsilon = 0
> > >
> > > Thanks for the clarification. To avoid the confusion, changing the sentence before the equation to the below would help.
> > > "the influence of a training point (xm, ym) on the parameter is _defined_ as"
> > >
> > >
> > > > > "I do not recall Martens making this argument, and it is not clear what is meant by "low-noise distribution".
> > > >
> > > > Thank you for this comment. We changed the reference to Kunstner et al. Limitations of the empirical fisher approximation for natural gradient descent; we also changed "low-noise" to "realizable", the term they use in Section 4.2. By low-noise we mean a model that converged to $p(y|x) \approx 1$ on the training data, which is better described as realizable.
> > >
> > > The change to "realisable" clears this concern.
> > > My issue was that "low-noise" could also describe settings where we know the empirical Fisher does not hold, for example if the model can interpolate the data in a regression setting (eg given some $x_1, ..., x_n$ and $y_i = \langle x_i, w\rangle$ for some $w$, and a linear model $f(w) = \||Xw-y\||^2$, for which the gradients are all 0 at the minimum but the Hessian is non-zero, the criticism raised by Kunstner et al. at the end of section 4.2).

---

> ### Author Response · Authors · 2024-11-28
> **Response to Reviewer qCYo**
>
> Thank you for clarifying your concern further. We agree with the reviewer that it is important to explicitly state that the algorithm does not really converge with a constant learning rate.
>
> Our observation is that in practice the sampling error appears to be not as important. In case of large models, where $\eta$ is already small, the corresponding sampling error can be negligible. Although technically the algorithm does not converge, it can provide sufficient approximation.
>
> We do agree with the reviewer that our choice of hyper parameters should be stated as heuristic.
>
> Below we suggest the following evaluations for some models we mention in the paper:
>
> |  | | $\| u_{T} - \hat{u}^{\star}\|^2$ |  sampling error   | $\|\hat{u}^{\star}\|^2$ |
> |----------|----------|----------|----------|----------|
> | OPT | 1.3B | 0.18 | 0.12 | 10.8 |
> | Llama-1 | 7B | 0.61 | 0.04 | 20.3 |
> | Mistral | 7B | 2.77 | 0.86 | $3.7 \times 10^4$ |
>
> Here, $ u_T $ is computed according to the hyperparmeters in Table 1, and we evaluate $ \hat{u}^{*} $
> by running LiSSA with a 2x smaller learning rate, a 4x larger batch size, and a 4x larger T. Sampling error is evaluated proportionally to $ g^T$ $\hat{u} $.
>
> Upon reviewing the sampling error, we found a mistake. In Theorem 1, the term $ \eta^2 \frac{Tr(H)}{|B|} g^T(H + \lambda)^{-1} g$ has to be replaced with $ \frac{\eta Tr(H)}{\lambda |B|} g^T(H + \lambda)^{-1} g$. The proof stands the same, we only forgot to multiply the error term $ \| \tilde{\Delta}\|^{2} $ by $\delta^{-1}$ when applying Lemma 2 at the end of the proof of Theorem 1.
>
> We will make the corresponding changes in case we get the chance to update the submission. To make the distinction between "convergence" and "approximation", e.g. replacing *"we find that the batch size has to be sufficiently large for LiSSA to converge"* will be replaced with *"we find that  the
> batch size must be sufficiently large for reliable approximation of iHVP"* in the abstract; furthermore, we will emphasise this difference in the passage directly after Theorem 1, perhaps including the evaluations above.
>
> Regarding the optimality of the choice of learning rate, we want to emphasize the paper does not actually that it is optimal, we refer to the previous work for this choice. We agree that in order to avoid misleading the reader, we should make it explicit that this choice is not necessarily optimal and other choices might work as well.
>
> Regarding the reference to Ma, Bassily, Belkin, we will remove the part about critical batch size. Indeed, their definition of critical batch size is different. However, there is similarity. In our result, we show that batch size |B| >= m^{*} is sufficient for learning in $O(\lambda_{\max} / \lambda)$ steps. If we double the batch size, the sampling error decreases, but we can no longer proportionally increase the learning rate. Therefore, we still need to conduct the same number of steps due to the convergence error. This example corresponds to their "saturation" regime.
>
> Thanks a lot you for your efforts and valuable feedback, we really appreciate it.

---

> > ### Comment · Reviewer_qCYo · 2024-11-28
> >
> > Thank you for your response. The above phrasing is more appropriate and would address my main concern regarding overstatement/acknowledgement of the optimization literature. I have increased my score.

---

### Official Review · Reviewer_XESR · 2024-11-03

**Soundness:** 2
**Presentation:** 3
**Contribution:** 2
**Rating:** 5
**Confidence:** 3

**Summary:**

Influence function captures how the optimal model weights change with respect to a newly added data point and has the form of an inverse Hessian-vector product. LiSSA is classical solver to compute the influence function which essentially applying stochastic gradient descent to a quadratic objective. This paper proposes conditions to choose the hyperparameters (proximal hyperparameter, batch size, step size) of the LiSSA algorithm for computing the influence functions of a model to a newly added data point. The conditions guarantee the convergence of LiSSA in expectation. The convergence analysis shows that the batch size has to be sufficiently large in order for the LiSSA algorithm to converge.

**Strengths:**

- The paper is well-organized and clearly written
- This paper gives the conditions on the hyperparameters for the LiSSA algorithm to converge. Such results are missing in the current literature.

**Weaknesses:**

- The contribution is somewhat limited because the convergence analysis seems to be standard, and the algorithm is only for computing the influence function.
- Some results in section 4 (Figure 1) seem to indicate that, in some situations, the influence function is not a good indicator of how the sensitive the optimal weights are with respect to newly added data points, as the LiSSA influence and PBRF influences are not correlated at all.
- The paper can be strengthened by reviewing more related work. E.g., are there other related algorithms for computing the influence function? Are there other variants of the influence function? Other work that analyze inverse Hessian-vector products (which are listed in the conclusion)? This can show the paper's position within the broader literature

**Questions:**

- In section 5, the paper claims that the idea of inverse Hessian makes sense because small eigenvalues correspond to more specific and informative content. However, in traditional interpretation, small eigenvalues correspond to noise and carry little to no information. What is it not the case here? Moreover, are there references for the claim that large eigenvalues correspond to general language structure?
- The Hutchinson estimation and randomized sketching are used to estimate Tr(H) and lambda_max. In the experiments, how many Hessian-vector products are needed to get a good estimation?
- In Figure 3, lambda=5 is used, which seems to be large given that the GNH matrix in (7) is already SPSD. How does the results change using a smaller lambda?

---

> ### Author Response · Authors · 2024-11-21
> **Response to reviewer XESR**
>
> We thank the reviewer for valuable feedback. Below we address some of the comments in the review
>
> 1) *"The contribution is somewhat limited because the convergence analysis seems to be standard, and the algorithm is only for computing the influence function."*
>
> We point out that the convergence analysis is slightly tighter than in more general results who rely on a weaker bound on $\mathbf{E} || \tilde{H}_t ||^{2}$. The closest result to ours is the analysis in Ma, Bassily, Belkin Interpolation...:, who consider the linear regression case. Although they also address the question of critical batch size, they do not make a connection to $Tr(H)$, rather rely on a uniform bound on the feature norm (in our case it is equivalent to norm of the gradient). We also note that our counter example showing that batch size has to be sufficiently large, accounts for arbitrary $H$, unlike Ma, Bassily and Belkin who only consider $H = cI$. Naturally, this is thanks to our condition C.1. In the revised version we put more emphasis on explaining why we believe it is a reasonable assumption, and we also add additional validation experiment.
>
> 2) *"Some results in section 4 (Figure 1) seem to indicate that, in some situations, the influence function is not a good indicator of how the sensitive the optimal weights are with respect to newly added data points, as the LiSSA influence and PBRF influences are not correlated at all." *
>
> We elaborate why this may happen in lines 385-388: either both influences are small, in which case they both approximate 0, or both influences are large, in which case the local Taylor approximation of PBO may not take place.
>
> 3) *"The paper can be strengthened by reviewing more related work. E.g., are there other related algorithms for computing the influence function? Are there other variants of the influence function? Other work that analyze inverse Hessian-vector products (which are listed in the conclusion)? This can show the paper's position within the broader literature"*
>
> We added some relevant literature in the revised version.
>
> 4) *"In section 5, the paper claims that the idea of inverse Hessian makes sense because small eigenvalues correspond to more specific and informative content. However, in traditional interpretation, small eigenvalues correspond to noise and carry little to no information. What is it not the case here? Moreover, are there references for the claim that large eigenvalues correspond to general language structure?"*
>
> Thank you for pointing this out. For better clarity, we change "small" to "remaining", that is not from the top of the spectrum.
>
> Higher information is traditionally associated with something that is less likely to observe. We could think of top eigenvectors as something that is more likely to observe. We also make a comparison to popular TF-IDF index in lines 482-487, where more frequent words are weighted less, i.e. bearing less information.
>
> Perhaps the following argument can also help the reviewer agree with us. The information may be interpreted as something that helps the model learning, especially given that we are talking about the distribution of the gradients. In the early deep learning work, such as Martens. Deep learning via Hessian-free optimization and Sutskever et al. On the importance of momentum and initialization, the authors argue that it is more efficient to train in low-curvature directions ("valley") where the gradients are more persistent along the training path.
>
> "The Hutchinson estimation and randomized sketching are used to estimate Tr(H) and lambda_max. In the experiments, how many Hessian-vector products are needed to get a good estimation?"
>
> For Hutchinson estimation, it is straighforward to evaluate std, which we give in Table 2. For evaluation of lambda_max with sketching we do not have a way to calculate std. The bound in Swartworth and Woodruff depends on Tr(H^2). Based on our evaluation of Tr(H^2), we suggest to take projection dimension 5000, see lines 968-971
>
> 5) *"In Figure 3, lambda=5 is used, which seems to be large given that the GNH matrix in (7) is already SPSD. How does the results change using a smaller lambda?"*
>
> Typically the Hessian itself is nearly singular, for example, some analysis is available in Schioppa (2024) Gradient Sketches for Training Data Attribution and Studying the Loss Landscape. The value of lambda directly affects the speed of convergence, the lower the value $1 - \lambda\eta$, the better. We choose lambda = 5 so we can take a relatively adequate number of steps. Notice that the top eigenvalue is much larger than 5 (780 for OPT, Table 2).

---

> > ### Comment · Reviewer_XESR · 2024-11-26
> > **Response**
> >
> > I thank the authors for their response. I think they have addressed my concerns. After reading the comments and discussions between the authors other reviewers. I will keep my score as is.

---

### Author Response · Authors · 2024-11-25
**Revision summary**

We thank the reviewers for taking the time to read the paper and write extensive high-quality reviews. We are grateful for comments and suggestions which helped us to revise the paper.

We summarise the changes below:

1. As was rightfully pointed out by the reviewers, the analysis of the error of the approximation is not new. We make this clear in the revised version and put more emphasis on condition C.1 which appears to be more novel compared to the existing literature. Firstly, we now make a precise theoretical statement in form of Lemma 1 in the main text.

2. We also added an additional experiment, where rather than comparing traces of LHS and RHS in condition C.1, we evaluate the spectrum of a low-dimensional random projection of their difference. This experiment is rather expensive, we only conduct it for a small GPT2, which however includes the case of dependent sampling from Lemma 1.

We additionally want to draw reviewers' attention to our counter-example, Lemma 3 in the appendix (revised accordingly to changes in the main text). It confirms that the requirement for the batch size is necessary, with the counter-example satisfying condition C.1 as well. Furthermore, we also want to remind the reviewers about the experiment in Figure 2 which supports the batch size requirement empirically. This kind of results generally do not appear in relevant optimization literature, with exception of lower bound in Ma, Bassily, Belkin, who only cover the case where $H = c I$.

We really appreciate the time and effort it takes to produce the reviews. Since the discussion is ending soon, we hope the reviewers could take some extra time to comment if the questions in their original reviews were not addressed properly. Thanks!

---

### Meta-Review · Area_Chair_cmoo · 2024-12-19

**Metareview:**

This paper analyzes the convergence for LiSSA algorithm (used to compute inverse Hessian-vector products, which are useful for computing influence functions). The results show that a large (but mild) batch is required for convergence, and the optimal hyperparameters for the algorithm depends on spectral properties of the Hessian  (trace and largest eigenvalue). The paper then proposes to estimate these values by sketching method and give a practical way of finding the hyperparameters. The reviewers generally agree that this way of finding hyperparameters is interesting and practical. There are some concerns about whether this is only for a specific problem of estimating the influence function, or the novelty of the theoretical results. The author response clarified many aspects of the paper. However a couple reviewers are still not convinced.

**Additional Comments On Reviewer Discussion:**

The reviewers have read the author response and changed their recommendations accordingly. Some reviewers remain skeptical about the significance of the result.

---

### Decision · Program_Chairs · 2025-01-22

Reject